# A Simple Image Segmentation Framework via In-Context Examples

**Yang Liu**[1], **Chenchen Jing**[1], **Hengtao Li**[1], **Muzhi Zhu**[1]
**Hao Chen**[1]*, **Xinlong Wang**[3], **Chunhua Shen**[1,2]

[1]Zhejiang University, China [2]Ant Group [3]Beijing Academy of Artificial Intelligence

## Abstract

Recently, there have been explorations of generalist segmentation models that can effectively tackle a variety of image segmentation tasks within a unified in-context learning framework. However, these methods still struggle with task ambiguity in in-context segmentation, as not all in-context examples can accurately convey the task information. In order to address this issue, we present SINE, a simple image **S**egmentation framework utilizing **in**-context **e**xamples. Our approach leverages a Transformer encoder-decoder structure, where the encoder provides high-quality image representations, and the decoder is designed to yield multiple task-specific output masks to eliminate task ambiguity effectively. Specifically, we introduce an In-context Interaction module to complement in-context information and produce correlations between the target image and the in-context example and a Matching Transformer that uses fixed matching and a Hungarian algorithm to eliminate differences between different tasks. In addition, we have further perfected the current evaluation system for in-context image segmentation, aiming to facilitate a holistic appraisal of these models. Experiments on various segmentation tasks show the effectiveness of the proposed method.
Our code is released at: https://github.com/aim-uofa/SINE

## 1 Introduction

Image segmentation [65, 31, 24, 58, 46] involves localizing and organizing concepts at the pixel level. Different definitions of concepts, such as foreground, category, and object instance, lead to different types of segmentation tasks. Recent years, we have witnessed great progress in developing more accurate and faster algorithms for various segmentation tasks such as semantic segmentation [35, 64, 48], instance segmentation [19, 7, 2], panoptic segmentation [24, 4, 5], foreground segmentation [52], interactive segmentation [57, 37, 25]. Nonetheless, most existing segmentation methods are tailored for certain tasks and cannot be applied to other tasks.

More recently, a few works [54, 56] have explored generalist segmentation models that are capable of solving diverse and unlimited segmentation tasks via in-context learning [3]. Painter [54] performs in-context training with masked image modeling and can achieve various tasks according to the in-context visual prompts. SegGPT [56] focuses on visual segmentation and introduces in-context segmentation, which unifies multiple segmentation tasks by incorporating both a target image and an annotated reference image as input. To encourage the model to leverage contextual information, SegGPT employs a random coloring scheme during the task completion process. However, these models still struggle with task ambiguity in in-context segmentation.

As shown in Figure 1(a), the in-context segmentation model needs to understand the task and content information conveyed by the in-context example and segments related concepts on the target image.

---

*HC is the corresponding author.

38th Conference on Neural Information Processing Systems (NeurIPS 2024).

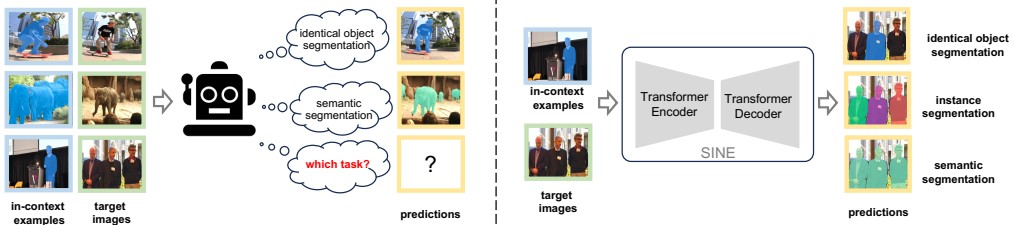

(a) Ambiguity in traditional in-context segmentation framework.  (b) An illustration of the of in-context segmentation framework of SINE.

**Figure 1** – Illustration of ambiguity in traditional in-context segmentation framework and an overview of our SINE framework.

However, not all in-context examples can convey the task information accurately. For example, when presented with a photo of a particular individual for segmenting the target image that includes him/her and others, which visual concepts should be segmented? Specifically, should it be limited to the individual alone, encompass the instance segmentation of all persons, or focus on semantic segmentation? Whether the pixels of the target image need to be segmented depends on their similarity to the in-context example. Ambiguous in-context examples can make it difficult for traditional in-context segmentation models to clearly define the boundaries between different tasks, resulting in undesired outputs.

To address this issue, we present SINE, a simple image **S**egmentation framework via **IN**-context **E**xamples. Drawing inspiration from the segment anything model [25], which addresses point ambiguity by generating multiple masks of different granularity, SINE predicts multiple output masks, custom-made for tasks of varying complexity. This ranges from identical objects, instances to overall semantic conception. SINE unifies existing segmentation tasks with various granularities, intending to achieve a broader task generalization. SINE leverages a Transformer encoder-decoder structure. The encoder contains a frozen pre-trained image encoder offering high-quality image representations and an In-context Interaction module to complement in-context information and learn correlations between the target and the reference image features. We propose a novel Matching Transformer (M-Former) for efficient multiple-task decoding. M-Former is implemented by a dual-path Transformer. One path is used for information interaction between object queries with image features. The second path is employed to enhance the semantic prototypes for accurate matching. In addition, a fixed matching and a Hungarian algorithm are used to eliminate differences between different tasks.

SINE achieves impressive performance with fewer trainable parameters compared with SegGPT. Our qualitative results demonstrate that SINE can effectively address the problem of task ambiguity in in-context segmentation, while SegGPT only outputs the semantic segmentation results. SINE achieves state-of-the-art or competitive performance on existing in-context image segmentation benchmarks [56], including few-shot semantic segmentation, video object segmentation. In addition, we further introduce few-shot instance segmentation to the current evaluation system for facilitating a holistic appraisal of these models. SINE provides a baseline result for in-context segmentation models to promote the development of this field. Finally, comprehensive ablation studies verify the effectiveness of the proposed components. Our main contributions are as follows:

- To our knowledge, our method is the first to investigate the task ambiguity of in-context segmentation, and we present a simple but effective framework to address the issue.
- We introduce a Matching Transformer to unleash the potential of frozen pre-trained image models on various segmentation tasks with a low training cost.
- Our comprehensive results demonstrate that SINE can address a broad range of segmentation tasks, including few-shot semantic segmentation, few-shot instance segmentation, video object segmentation. Ablation studies show the effectiveness of the proposed components.

## 1.1 Related Work

**Image Segmentation** Segmentation involves localizing and organizing meaningful concepts at the pixel level. Different definitions of concepts, such as foreground, category, and object instance, lead to different types of segmentation tasks. Specifically, semantic segmentation [65] requires

semantic classification at the pixel level, while instance segmentation [31] is to identify and localize various object instances. Panoptic Segmentation [24] introduces a more challenging task by unifying semantic segmentation and instance segmentation. Video object segmentation [58, 46] is to segment an identical object throughout a video sequence. Most existing segmentation methods are tailored for certain tasks and cannot be applied to other tasks.

Traditional segmentation methods [35, 29, 64, 48, 19, 7, 2, 24, 4, 5, 10, 66] have been designed for identical segmentation tasks and trained on a specific dataset. These methods have limited generalization when transferring to other datasets or tasks. To solve this challenge, we unify various segmentation tasks into in-context segmentation [56, 34, 27, 39] and introduces a simple visual segmentation framework via in-context examples. Prompted by a given in-context example, our method can perform few-shot segmentation across various datasets and tasks.

**In-Context Visual Learning** Recently, in-context learning has been successfully used in NLP tasks and vision-and-language tasks. Bar *et al.* [1] first introduced in-context learning in computer vision. They cast the in-context learning as an image inpainting problem, where models need to fill in a hole in a concatenated image containing several examples and a new input image. They demonstrate the effectiveness of the strategy for foreground segmentation, single object detection, and colorization. Painter [54] adopts masked image modeling on continuous pixels to perform in-context training with supervised datasets, and achieves competitive results on seven vision tasks. SegGPT [56] focuses on the segmentation task and uses random coloring scheme that forces the model to reference contextual information to complete the assigned task. DiffewS [67] effectively leverages in-context visual learning to unleash the potential of Stable Diffusion [49] in few-shot semantic segmentation.

Our work focuses on segmentation tasks. These aforementioned models struggle with the task ambiguity in in-context segmentation. By contrast, our SINE can effectively address the problem of task ambiguity in in-context segmentation by disentangling the specific task from the in-context example and understanding the semantic concepts of the prompts to output results at different levels of task granularity from the identical object, instance, to semantics.

## 2  Preliminary

In this section, we first formulate the problem setting of in-context segmentation. Then, we revisit SegGPT, the previous in-context segmentation model.

### 2.1  Problem Formulation

In-context segmentation aims to identify a specific task and objects within the given in-context example, including a reference image $\mathbf{x}_r$ and its annotations $\mathbf{y}_r = \{(m_r^i, c_r^i)\}_{i=1}^{N}$, and segment the interested objects in the target image $\mathbf{x}_t$. $m_r^i$ denotes the $i$-th reference mask and $c_r^i$ denotes its class label, $N$ denotes the number of the reference masks. The interested objects are related to the in-context example, which can be an identical object for video object segmentation or all objects of the same semantic concept for instance segmentation and semantic segmentation. We mainly focus on three semantic granularities in the task ambiguity, *i.e.*, identical object (ID), instance, and semantic. The identical object segmentation can be seen as finer granularity instance segmentation, and instance segmentation can be transformed into semantic segmentation by merging instance masks belonging to the same category. Based on this relationship between these tasks, we unify them into instance segmentation.

### 2.2  A Revisit of SegGPT

SegGPT [56] introduces in-context segmentation, which incorporates both the images and mask annotations into an RGB image to convey the specific task to be performed and identify the objects to be segmented simultaneously. SegGPT takes stitched reference and target images as input and employs the masked image modeling algorithm [18] and smooth-$\ell 1$ [15] loss to train a Vision Transformer [8] encoder. During inference, SegGPT enables segmenting everything via an in-context example, including a reference image and its annotation image. Given a target image, it is stitched with the reference image and fed into SegGPT to get the corresponding in-context predictions. Although SegGPT has achieved great success in various segmentation tasks, it faces two challenges: 1) In many cases, in-context examples may not accurately convey the task information. For example,

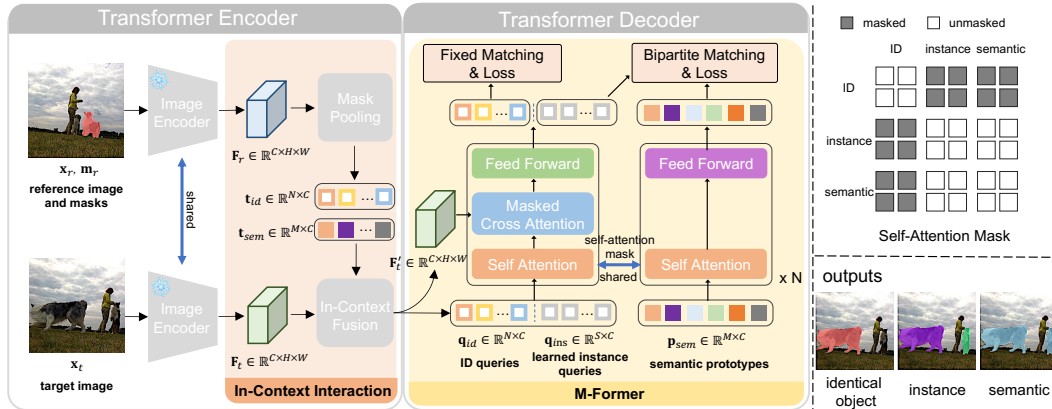

**Figure 2** – An overview of SINE. SINE is a Transformer encoder-decoder structure, including a frozen pre-trained image encoder, an In-Context Interaction module, and a lightweight Matching Transformer (M-Former) decoder. The top right corner is the self-attention mask of M-Former. The down right corner shows the different task outputs, from identical object, instance, to semantic.

when an in-context example only consists of a single object and its annotation, the lack of additional task-related information can lead to incorrect outputs. 2) SegGPT utilizes a single Transformer encoder structure for both feature extraction and task-specific decoding purposes. This will introduce complexity in the in-context segmentation process and lead to a sub-optimal solution.

Inspired by the segment anything model [25], which solves point ambiguity by outputting multiple different granularity masks simultaneously, we focus solely on the various content of in-context examples and endow the model with the ability to predict multiple output masks for different tasks. In addition, we disentangle the function of the Transformer encoder within SegGPT and deploy a Transformer encoder and decoder to perform feature extraction and task decoding, respectively.

## 3 Method

We present SINE, a simple image **S**egmentation framework via **IN**-context **E**xamples. SINE can effectively address the problem of task ambiguity in in-context segmentation by disentangling the specific task from the in-context example and understanding the semantic concepts of the prompts to output results at different levels of task granularity from the identical object, instance, to semantics. We elaborate on our method in the following subsections.

### 3.1 Overview

The overview of the proposed SINE framework is depicted in Figure 2. We build SINE based on the classic Transformer structure [53, 8] and introduce some effective designs targeted for the in-context segmentation task, including a frozen pre-trained image encoder, an In-Context Interaction module, and a lightweight Matching Transformer (M-Former) decoder.

We use the frozen pre-trained image encoder to encode $\mathbf{x}_r$ and $\mathbf{x}_t$, resulting in the reference feature $\mathbf{F}_r \in \mathbb{R}^{C \times H \times W}$ and target feature $\mathbf{F}_t \in \mathbb{R}^{C \times H \times W}$, where $H$, $W$, and $C$ denote the image features' height, width and the number of channels, respectively. Inspired by the in-context learning in NLP [3], we enable SINE to grasp the in-context correlations that coexist between the reference and the target images. Specifically, we develop an In-Context Interaction module, a component tailored to capture the semantic correlations between $\mathbf{F}_r$ and $\mathbf{F}_t$ and outputs the enhanced target feature $\mathbf{F}_t^{'} \in \mathbb{R}^{C \times H \times W}$, the ID queries $\mathbf{q}_{id} \in \mathbb{R}^{N \times C}$ and the semantic prototypes $\mathbf{p}_{sem} \in \mathbb{R}^{M \times C}$.

SINE is built as a query-based segmentation model, following DETR [4] and Mask2Former [5]. We employ the ID queries $\mathbf{q}_{id}$ to identify and locate objects within the target image that have identical counterparts within the reference image. Furthermore, we apply learnable instance queries $\mathbf{q}_{ins} \in \mathbb{R}^{S \times C}$ to identify and locate objects within the target image that share the same semantic labels in the reference image. The M-Fomer decoder is employed to update these queries and

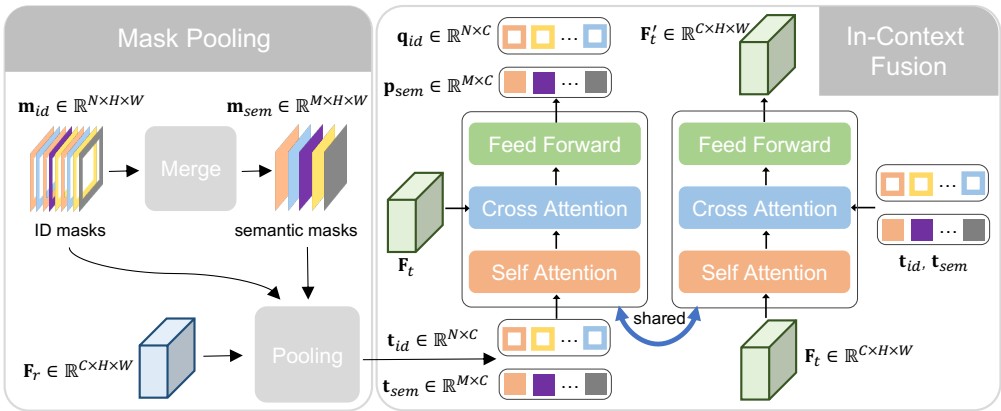

**Figure 3** – Illustration of the In-Context Interaction module. This module aims to complement in-context information between reference and target. The ID and semantic tokens are extracted by the Mask Pooling. The enhanced target feature, the ID queries, and the semantic prototypes are outputted by the In-Context Fusion module.

prototypes. Then, a prediction feed-forward network [4] is employed to predict ID and instance outputs, respectively.

### 3.2 In-Context Interaction

The purpose of In-Context Interaction is to complement in-context information and produce correlations between reference and target image features. As illustrated in Figure 3, we use a Mask Pooling process to extract the ID and semantic tokens, represented as $\mathbf{t}_{id} \in \mathbb{R}^{N \times C}$ and $\mathbf{t}_{sem} \in \mathbb{R}^{M \times C}$. Specifically, we transfer $\mathbf{m}_r$ into ID masks $\mathbf{m}_{id} \in \mathbb{R}^{N \times H \times W}$ by assigning different ID labels for each mask and semantic masks $\mathbf{m}_{sem} \in \mathbb{R}^{M \times H \times W}$ by merging the masks with same category label, where $N$ and $M$ are the numbers of ID and semantic masks. Then, we use these masks to pool over the reference feature $\mathbf{F}_r$ and obtain $\mathbf{t}_{id}$ and $\mathbf{t}_{sem}$, the pooling process similar to [14].

We introduce an In-Context Fusion module to enable the in-context correlation between the reference and target features. The process of this module can be summarized as follows:

$$\left\langle \mathbf{q}_{id}, \mathbf{p}_{sem}, \mathbf{F}_t' \right\rangle = InContextFusion\left(\mathbf{t}_{id}, \mathbf{t}_{sem}, \mathbf{F}_t; \theta\right),\tag{1}$$

where $\theta$ is the parameters of the In-Context Fusion. The module is a Transformer block [53, 4] including a self-attention, cross-attention, and a feed-forward network. The tokens ($\mathbf{t}_{id}$ and $\mathbf{t}_{sem}$) and target feature ($\mathbf{F}_t$) are fused by this shared module, where they are used as keys and values for each other in the cross-attention, and the enhanced target feature $\mathbf{F}_t'$, the ID queries $\mathbf{q}_{id}$ and the semantic prototypes $\mathbf{p}_{sem}$ can be obtained.

### 3.3 Matching Transformer

M-Former aims to decode the enhanced target feature into different task outputs, from identical object, instance, to semantic, and enables comprehensive and efficient performance for in-context segmentation. Thus, in addition to the ID queries $\mathbf{q}_{id}$ and the semantic prototypes $\mathbf{p}_{sem}$, we also incorporate a set of learnable instance queries $\mathbf{q}_{ins}$ for predicting instances. To perform in-context segmentation and eliminate task ambiguity effectively, the design of M-Former needs to consider 1) the semantic prototypes need to assign semantic information to the instance queries, 2) avoiding coarse-grained semantic prototypes to contaminate fine-grained ID queries, and 3) the queries interact with the target feature.

Encouraged by the above analyses, the M-Former is implemented by a dual-path Transformer decoder sharing the self-attention layers. One path is utilized to interact $\mathbf{F}_t'$ and queries ($\mathbf{q}_{id}$ and $\mathbf{q}_{ins}$) for extracting correlated feature with the in-context example from the target image. This path consists of a series of self-attention, masked cross-attention [5], and feed-forward network. The second path is employed to enhance the semantic prototypes $\mathbf{p}_{sem}$ for a more accurate matching. The two paths

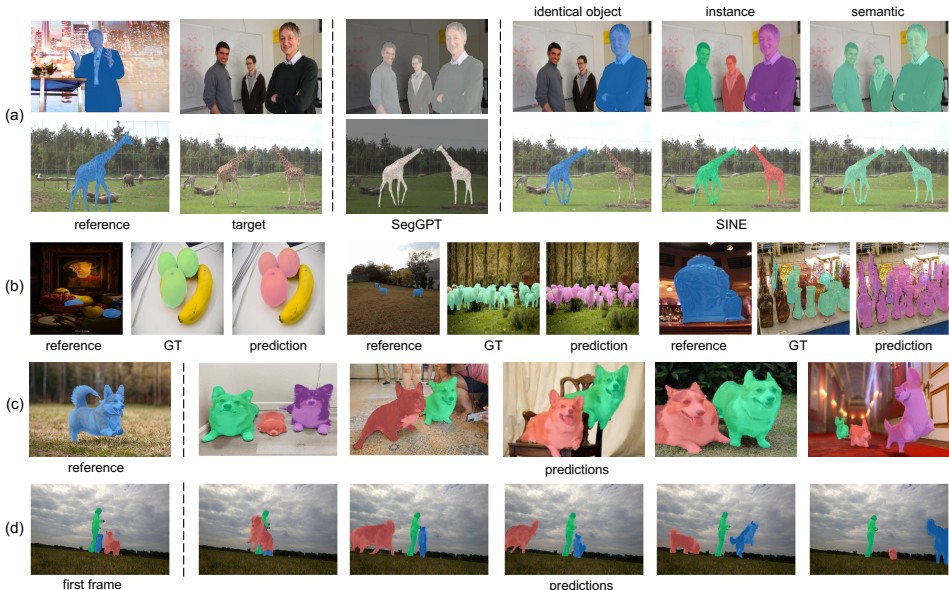

**Figure 4** – Qualitative results of SINE. (a) Comparison between SegGPT and SINE for addressing ambiguity in in-context segmentation. (b) Few-shot semantic segmentation. (c) Few-shot instance segmentation. (d) Video object segmentation.

share the self-attention layers for assign semantics from $\mathbf{p}_{sem}$ to $\mathbf{q}_{ins}$. To avoid semantic prototypes contaminating fine-grained ID queries, we apply a self-attention mask within the shared self-attention layers as shown in at the top right corner of Figure 2. M-Former has N blocks. The process of M-Former can be summarized as follows:

$$\left\langle \mathbf{q}_{id}^l, \mathbf{q}_{ins}^l, \mathbf{p}_{sem}^l \right\rangle = MFormer_l \left( \mathbf{q}_{id}^{l-1}, \mathbf{q}_{ins}^{l-1}, \mathbf{p}_{sem}^{l-1}; \theta^l, \mathbf{F}_t^{'} \right), \tag{2}$$

where $l$ denotes the layer number of M-Former and $\theta^l$ is the parameters of the $l$-th M-Former block. Then, a prediction feed-forward network is employed to predict ID and instance outputs, respectively.

For instance segmentation, we use the updated semantic prototypes $\mathbf{p}_{sem}$ as the classifier and let $\hat{\mathbf{y}}_{ins} = \{\hat{y}_{ins}^i\}_{i=1}^{S}$ denote the set of $S$ instance predictions. The ground truth is denoted by $\mathbf{y}$. We use the Hungarian loss [4, 5] to learn SINE. Specifically, we compute the assignment costs between prediction $\hat{y}_{ins}^i$ and ground truth $y^j$ for the matching problem, *i.e.*, $-p_i(c^j) + \mathcal{L}_{\text{mask}}(\hat{m}_{ins}^i, m^j)$, where $(c^j, m^j)$ is the class and mask of the ground truth object, $c^j$ may be $\varnothing$. $p_i(c^j)$ is the probability of class $c^j$ for $i$-th instance query, $\hat{m}_{ins}^i$ denotes its predicted mask. $\mathcal{L}_{\text{mask}}$ is a binary mask loss and Dice loss [40]. The Hungarian loss is

$$\mathcal{L}_{\text{Hungarian}}(\hat{\mathbf{y}}_{ins}, \mathbf{y}) = \sum_{j=1}^{S} \left[ -\log p_{\sigma(j)}(c^j) + \mathbb{1}_{c^j \neq \varnothing} \mathcal{L}_{\text{mask}}(\hat{m}_{ins}^{\sigma(j)}, m^j) \right], \tag{3}$$

where $\sigma(j)$ denotes the resulting index of the bipartite matching. To endow SINE with the ability to predict an identical object, we use the different cropped views of the same instance within the image as reference-target image pairs. Let $\hat{\mathbf{y}}_{id} = \{\hat{y}_{id}^i\}_{i=1}^{N}$ denote the set of $N$ ID predictions. Because the relationship between reference and target IDs is fixed and can be accurately determined, we perform fixed matching between the predictions and the ground truth. The loss can be written as

$$\mathcal{L}_{\text{ID}}(\hat{\mathbf{y}}_{id}, \mathbf{y}) = \sum_{i=1}^{N} \left[ -\log p_i(c^i) + \mathbb{1}_{c^i \neq \varnothing} \mathcal{L}_{\text{mask}}(\hat{m}_{id}^i, m^i) \right], \tag{4}$$

where $(c^i, m^i)$ is the ground truth class and mask, $c^i \in \{1, \varnothing\}$, $c^i = 1$ denotes that an object appears in both reference and target images simultaneously. The total loss is $\mathcal{L} = \mathcal{L}_{\text{Hungarian}} + \mathcal{L}_{\text{ID}}$. Once the training is finished, the full capability of SINE is unleashed during inference. SINE can address the ambiguity within the in-context examples and output predictions for different segmentation tasks.

| Methods | Venue | COCO-20$^i$ | | PASCAL-5$^i$ | | LVIS-92$^i$ | |
|---|---|---|---|---|---|---|---|
| | | one-shot | few-shot | one-shot | few-shot | one-shot | few-shot |
| *specialist model* | | | | | | | |
| HSNet [41] | ICCV'21 | 41.7* | 50.7* | 68.7* | 73.8* | 17.4 | 22.9 |
| VAT [20] | ECCV'22 | 42.9* | 49.4* | 72.4* | 76.3* | 18.5 | 22.7 |
| FPTrans [62] | NeurIPS'22 | 56.5* | 65.5* | 77.7* | 83.2* | - | - |
| *generalist model* | | | | | | | |
| Painter [54] | CVPR'23 | 32.8 | 32.6 | 64.5 | 64.6 | 10.5 | 10.9 |
| SegGPT [56] | ICCV'23 | 56.1 | 67.9 | 83.2 | 89.8 | 18.6 | 25.4 |
| PerSAM-F [63] | ICLR'24 | 23.5 | - | - | - | 18.4 | - |
| Matcher [34] | ICLR'24 | 52.7 | 60.7 | - | - | 33.0 | 40.0 |
| SINE | this work | 64.5 | 66.1 | 85.4 | 86.2 | 31.2 | 35.5 |

**Table 1** – Results of few-shot semantic segmentation on COCO-20$^i$, PASCAL-5$^i$, and LVIS-92$^i$. Gray indicates the model is trained by in-domain datasets. $*$ indicates that the categories in training cover the categories in testing within the same dataset.

## 4 Experiments

**Training Data** We train our model with a diverse set of segmentation datasets, including semantic, instance, and panoptic segmentation. Specifically, we utilize three visual perception datasets: **ADE20K** [65] is a popular semantic segmentation dataset, providing semantic labels for 150 categories. For panoptic segmentation, the 150 categories can be divided into 100 "things" and 50 "stuff" categories. It has 25K images, including 20K for training, 2K for validation, and 3K for testing. **COCO** [31] is a widely-used dataset that supports object detection, instance segmentation, and panoptic segmentation. It contains 80 "things" and 53 "stuff" categories, with 118K training and 5K validation images. **Objects365** [51] is a large-scale high-quality object detection dataset. It contains 365 categories, 638K images, and 10M bounding boxes. We extend instance segmentation annotations for Objects365 by using the Segment Anything Model [25]. We convert different data annotations into the form of instance segmentation for unified mixed data training.

**Training Details** Following [34], we deploy the frozen DINOv2 (ViT-L) [44] with 304M parameters as the image encoder and train SINE with only 19M trainable parameters. The In-Context Fusion has one block and M-Former has six blocks. The model size of SINE is comparable with SegGPT (307M). SINE has fewer trainable parameters, leading to more efficient training. We train SINE about 50K steps with 64 batch sizes. We use Adam [36] optimizer and employ $\beta_1 = 0.9, \beta_2 = 0.999$ for optimization. We use a linear learning rate scheduler with a base learning rate of $1e{-}4$ and a warmup of 100 steps. The weight decay is set to 0.05. For data augmentation, we use random horizontal flipping and the large-scale jittering (LSJ) [13] augmentation with a random scale sampled from range 0.1 to 2.0 followed by a fixed size crop to $896 \times 896$. More implementation details are provided in the Appendix B.

### 4.1 Qualitative Results

We demonstrate that our SINE framework can effectively address the ambiguity in the in-context examples. As shown in Figure 4(a), it is difficult to understand which tasks should be executed via the reference image with an annotation of a particular individual.SegGPT only outputs the semantic segmentation result. In contrast, SINE outputs multiple outputs to avoid task ambiguity. We further visualize the results of SINE on few-shot semantic segmentation, few-shot instance segmentation, and video object segmentation. SINE showcases its capability to deliver highly accurate predictions across diverse tasks while retaining exceptional flexibility in the task definition.

### 4.2 Few-shot Semantic Segmentation

**Datasets** We revisit the few-shot semantic segmentation task into two settings: in-domain (COCO-20$^i$ [43]) and out-domain (PASCAL-5$^i$ [50], LVIS-92$^i$ [34]), based on whether the dataset has been seen during training. COCO-20$^i$ divides the 80 classes of COCO into four cross-validation folds with 20 test classes and 60 training classes for each fold. Similarly, PASCAL-5$^i$ is built on PASCAL, including four cross-validation folds. LVIS-92$^i$ is a more challenging benchmark for evaluating the generalization of a model across datasets based on LVIS [16], including ten folds and 92 classes for each fold. We comprehensively verify the few-shot semantic segmentation performance of SINE on these datasets following the evaluation scheme of [41, 56, 34].

| Methods | Venue | Det. | | Segm. | |
|---|---|---|---|---|---|
| | | 1 | 5 | 1 | 5 |
| *specialist model* | | | | | |
| Meta R-CNN [59] | ICCV'19 | - | 3.5 | - | 2.8 |
| MTFA [12] | CVPR'21 | 2.5 | 6.6 | 2.7 | 6.6 |
| iMTFA | | 3.3 | 6.2 | 2.8 | 5.2 |
| Meta-DETR [61] | TPAMI'22 | - | 15.4 | - | 8.1 |
| RefT [17] | arXiv'23 | - | 15.0 | - | 14.2 |
| *generalist model* | | | | | |
| SINE | this work | 18.0 | 22.5 | 16.9 | 21.8 |

**Table 2** – Results (AP) of few-shot object detection and instance segmentation on COCO-NOVEL with $K = \{1, 5\}$.

| Methods | Venue | Det. | | Segm. | |
|---|---|---|---|---|---|
| | | AP | AP50 | AP | AP50 |
| *1-shot* | | | | | |
| FGN [11] | CVPR'20 | - | 30.8 | - | 16.2 |
| MTFA [12] | CVPR'21 | 10.0 | 21.7 | 9.5 | 19.3 |
| iMTFA | | 11.5 | 22.4 | 8.6 | 16.3 |
| SINE | this work | 35.9 | 51.9 | 27.6 | 47.6 |
| *5-shot* | | | | | |
| SINE | this work | 42.8 | 62.1 | 33.3 | 57.7 |

**Table 3** – Results (AP and AP50) of few-shot object detection and instance segmentation on COCO2VOC with $K = \{1, 5\}$.

**Results** As shown in Table 1, we compare the SINE with a variety of specialist and generalist segmentation models. For COCO-20$^i$ and PASCAL-5$^i$, because SINE is trained with all data of COCO, we report the performance of specialist models trained on in-domain categories (marked by $^*$) for a fair comparison. Although SINE and SegGPT are trained on the COCO dataset, SINE achieves significant advantages in one-shot performance with a simpler in-context learning framework on COCO-20$^i$. In addition, without specific training on the PASCAL dataset, SINE achieves better performance with one-shot and comparable performance with few-shot to SegGPT on PASCAL-5$^i$. On the more challenging dataset LVIS-92$^i$, SINE outperforms Painter, SegGPT, and PerSAM-F, demonstrating superior generalization and versatility. The performance of SINE on LVIS-92$^i$ is slightly weaker compared to Matcher, which benefits from the SAM [25] pre-trained on a large-scale segmentation dataset. SINE achieves competitive results independently, without relying on extensive segmentation data.

**Please note** that SINE aims to provide insights for the research community to build a simple baseline for in-context segmentation, instead of SOTA. Despite having been trained on 12 diverse segmentation datasets [56], SegGPT demonstrates a limited performance when applied to LVIS-92$^i$. This indicates a need for developing more effective data-centric learning approaches specifically tailored for in-context segmentation. SINE is the first to explore the utilization of Objects365 with automatic mask annotations. The efficient design of SINE enables the training on Objects365 at the instance level. This enables SINE to only use 19M training parameters, making the training budget much smaller than SegGPT (304M) or SAM (600M) in Matcher.

### 4.3 Few-shot Instance Segmentation

**Datasets** Like few-shot semantic segmentation, we evaluate the performance of SINE of few-shot instance segmentation on both settings: in-domain on COCO-NOVEL [23] and out-domain on COCO2VOC [9] settings. COCO-NOVEL includes 20 classes that intersect with VOC and 5k test images. We evaluate 1-shot and 5-shot instance segmentation performance on this dataset. The COCO2VOC dataset is used to evaluate the cross-dataset generalization ability by using reference samples of COCO to test the VOC test set. We report the mean results of 10 groups of different reference images generated by different random seed for both two datasets.

**Results** Table 2 shows the results of few-shot object detection and instance segmentation. SINE outperforms specialist methods by a large margin at both 1-shot and 5-shot settings. For the results of COCO2VOC in Table 3, SINE shows better generalization ability compared to specialist models.

**Please note** that the goal of this experiment is not to prove that SINE achieves better few-shot instance segmentation performance compared to specialist segmentation models. The existing in-context segmentation models, such as SegGPT, are unsuitable to perform instance segmentation. SegGPT needs to use a sliding window to traverse all grids to predict the objects. The post-processing is complicated and ineffective. SINE is the first in-context segmentation model that can address few-shot instance segmentation. We hope SINE can be a baseline for in-context segmentation models to promote the development of this field.

### 4.4 Video Object Segmentation

**Datasets** For video object segmentation (VOS), we focus on the semi-supervised setting where the object masks of the first frame are provided, and the model is responsible for segmenting the object

| Methods | Venue | DAVIS 2017 | | | DAVIS 2016 | | | YouTube-VOS 2018 | | | | |
|---|---|---|---|---|---|---|---|---|---|---|---|---|
| | | $J\&F$ | $J$ | $F$ | $J\&F$ | $J$ | $F$ | $G$ | $J_s$ | $F_s$ | $J_u$ | $F_u$ |
| *with video data* | | | | | | | | | | | | |
| AGAME [22] | CVPR'19 | 70.0 | 67.2 | 72.7 | - | - | - | 66.0 | 66.9 | - | 61.2 | - |
| AGSS [30] | ICCV'19 | 67.4 | 64.9 | 69.9 | - | - | - | 71.3 | 71.3 | 65.5 | 75.2 | 73.1 |
| AFB-URR [28] | NeurIPS'20 | 74.6 | 73.0 | 76.1 | - | - | - | 79.6 | 78.8 | 83.1 | 74.1 | 82.6 |
| AOT [60] | NeurIPS'21 | 85.4 | 82.4 | 88.4 | 92.0 | 90.7 | 93.3 | 84.5 | 84.3 | 89.3 | 77.9 | 86.4 |
| SWEM [32] | CVPR'22 | 84.3 | 81.2 | 87.4 | 91.3 | 89.9 | 92.6 | 82.8 | 82.4 | 86.9 | 77.1 | 85.0 |
| XMem [6] | ECCV'22 | 87.7 | 84.0 | 91.4 | 92.0 | 90.7 | 93.2 | 86.1 | 85.1 | 89.8 | 80.3 | 89.2 |
| *without video data* | | | | | | | | | | | | |
| Painter [54] | CVPR'23 | 34.6 | 28.5 | 40.8 | 70.3 | 69.6 | 70.9 | 24.1 | 27.6 | 35.8 | 14.3 | 18.7 |
| SegGPT [56] | ICCV'23 | 75.6 | 72.5 | 78.6 | 83.7 | 83.6 | 83.8 | 74.7 | 75.1 | 80.2 | 67.4 | 75.9 |
| SEEM [68] | NeurIPS'23 | 58.9 | 55.0 | 62.8 | - | - | - | 50.0 | 57.2 | 38.2 | 61.3 | 43.3 |
| DINOv [27] | CVPR'24 | 73.3 | 71.0 | 75.7 | - | - | - | 60.9 | 65.3 | 70.0 | 52.3 | 57.9 |
| PerSAM-F [63] | ICLR'24 | 76.1 | 74.9 | 79.7 | - | - | - | 54.4 | 53.9 | 56.4 | 50.7 | 56.6 |
| Matcher [34] | ICLR'24 | 79.5 | 76.5 | 82.6 | 86.1 | 85.2 | 86.7 | - | - | - | - | - |
| SINE | this work | 77.0 | 72.6 | 81.3 | 82.3 | 81.4 | 83.2 | 66.2 | 69.1 | 57.6 | 71.7 | 66.5 |

**Table 4** – Results of video object segmentation on DAVIS 2017, DAVIS 2016, and YouTube-VOS 2018. Gray indicates the model is trained on target datasets with video data. $G$ is the average score over the "seen" and "unseen" classes in YouTube-VOS 2018.

| Methods | Venue | mIoU |
|---|---|---|
| *specialist model* | | |
| FCN [35] | CVPR'15 | 29.4 |
| RefineNet [29] | CVPR'17 | 40.7 |
| DPT [48] | ICCV'21 | 49.2 |
| Mask2Former [5] | CVPR'22 | 57.7 |
| *generalist model* | | |
| Painter [54] | CVPR'23 | 49.9 |
| SegGPT [56] | ICCV'23 | 39.6 |
| SINE | this work | 54.1 |

**Table 5** – Transfer performance on ADE20K semantic segmentation.

| Methods | Venue | PQ | $PQ^{Th}$ | $PQ^{St}$ |
|---|---|---|---|---|
| *specialist model* | | | | |
| PanopticFPN [24] | CVPR'19 | 40.8 | 48.3 | 29.4 |
| SOLOv2 [55] | NeurIPS'20 | 42.1 | 49.6 | 30.7 |
| Mask2Former [5] | CVPR'22 | 57.8 | 64.2 | 48.1 |
| UViM [26] | NeurIPS'22 | 45.8 | - | - |
| *generalist model* | | | | |
| Painter [54] | CVPR'23 | 43.4 | - | - |
| SegGPT [56] | ICCV'23 | 34.4 | - | - |
| SINE | this work | 51.0 | 57.8 | 40.8 |

**Table 6** – Transfer performance on COCO panoptic segmentation.

in all subsequent frames. We evaluate SINE on three validation datasets, including DAVIS 2017 [46], DAVIS 2016 [45], and YouTube-VOS 2018 [58]. We use the $J$ score and the $F$ score to evaluate the performance.

**Details** To effectively perform SINE on the VOS task, we introduce a group of memory banks for each object, maintaining the intermediate predictions. We determine which frame to retain in the memory according to the classification and mask scores. Considering that objects are more likely to resemble those in adjacent frames, we apply a time-based decay ratio to the scores, gradually reducing its value. In addition, we store the reference image and mask in memory to solve the case where objects vanish and reappear.

**Results** Table 4 compares the performance of VOS between SINE and different methods trained with or without video data. Without video data, SINE achieves competitive performance compared with the models trained with video data on DAVIS 2017. In addition, SINE outperforms recent generalist segmentation methods, such as Painter, SEEM, DINOv and PerSAM-F, on YouTube-VOS 2018. These results demonstrate that SINE has the potential to address video tasks.

### 4.5 Transfer Learning Experiments

We investigate the performance of SINE when transferring to common segmentation tasks, such as ADE20K semantic segmentation and COCO panoptic segmentation. Unlike traditional pre-trained methods, *e.g.*, MAE [18], require to fine-tune all model parameters to downstream tasks. We verify that our method can be efficiently transferred to these tasks via parameter efficient fine-tuning (PEFT) methods [38]. Specifically, We deploy the popular PEFT method, LoRA [21], on the frozen image encoder and fix the original parameters to test the transfer ability of SINE. We select rank 32 as the default setting. We train semantic prototypes and LoRA parameters for specific segmentation tasks.

**Semantic Segmentation** According to Table 5, our method SINE achieves 54.1% mIoU on the ADE20K semantic segmentation benchmark, outperforming other in-context segmentation models like SegGPT which trains a dataset-specific prompt using related dataset annotations. In addition, with only a few trainable parameters, our method achieves better or comparable performance to specialist segmentation models. It is worth noting that, unlike Mask2Former [5], SINE does not use multi-scale features for better performance.

| Training Data | LVIS-92$^i$ mean mIoU | COCO-NOVEL AP$_{box}$ | AP$_{mask}$ | DAVIS 2017 J&F |
|---|---|---|---|---|
| COCO | 24.4 | 10.5 | 11.4 | 63.2 |
| + ADE20K | 25.5 | 19.2 | 18.2 | 68.8 |
| + Objects365 | 28.3 | 22.4 | 21.5 | 77.0 |

**(a)** Ablation study of training data.

| Frames | DAVIS 2017 | | | | | |
|---|---|---|---|---|---|---|
| | 1 | 2 | 4 | 6 | 8 | 10 |
| J&F | 70.9 | 76.2 | 76.7 | 77.0 | 77.0 | 76.0 |
| J | 66.1 | 71.5 | 71.9 | 72.6 | 72.7 | 71.6 |
| F | 75.8 | 81.0 | 81.4 | 81.3 | 81.3 | 80.3 |

**(b)** Effect of the number of frames for VOS.

| Fusion | Decoder | LVIS-92$^i$ mean mIoU | COCO-NOVEL AP$_{box}$ | AP$_{mask}$ | DAVIS 2017 J&F |
|---|---|---|---|---|---|
| | Mask2Fomer | 22.2 | 8.6 | 9.9 | 61.1 |
| ✓ | Mask2Fomer | 23.5 | 8.9 | 10.6 | 62.3 |
| ✓ | M-Former | 24.4 | 10.5 | 11.4 | 63.2 |

**(c)** Effect of proposed components.

| Rank | #Params(M) | mIoU | mACC |
|---|---|---|---|
| 4 | 0.63 | 49.1 | 63.6 |
| 8 | 1.22 | 50.4 | 65.0 |
| 16 | 2.40 | 52.4 | 66.7 |
| 32 | 4.75 | 54.1 | 68.0 |
| 64 | 9.48 | 54.4 | 68.8 |

**(d)** Ablation study of LoRA rank on ADE20K.

**Table 7** – Ablation study. We report the mIoU of fold0 on LVIS-92$^i$, AP$_{box}$ and AP$_{mask}$ of seed0 on COCO-NOVEL, and J&F on DAVIS 2017. Pink is the default setting.

**Panoptic Segmentation** Table 6 shows that SINE also significantly outperforms other generalist segmentation models on the COCO panoptic segmentation task, demonstrating that the proposed method can enable the learned Transformer decoder to apply to more complicated panoptic segmentation effectively. By fixing the overall model parameters and adding only a few of LoRA parameters, SINE achieves competitive performance with the best specialist segmentation model, Mask2Former.

## 4.6 Ablation Study

As shown in Table 7, we conduct ablation experiments on LVIS-92$^i$ for few-shot semantic segmentation, COCO-NOVEL dataset for few-shot instance segmentation, DAVIS 2017 for video object segmentation, and ADE20K for the semantic segmentation task, to thoroughly validate the effectiveness of the proposed components and the impact of training with different datasets. Unless specified, only fold0 and seed0 are used for the ablation on the few-shot semantic/instance segmentation.

**Effect of different training data.** We train SINE on different data sources (Table 7a). We validate that incorporating more diverse semantic segmentation data, such as ADE20K, helps improve the model. Furthermore, we demonstrate for the first time that adding additional detection data, such as Object365, greatly enhances the model's in-context segmentation capability.

**Ablation study of proposed components.** Table 7c shows the impact of the proposed components and all experiments only use COCO as training data. The proposed In-Context Fusion module leads to improvements in mean mIoU, AP and J&F. Further enhancements are observed across all evaluation metrics when the M-Former is used. This demonstrates the effectiveness of the proposed components in improving the model's performance for various segmentation tasks.

**Varying the number of frames** Table 7b demonstrates a gradual improvement in the overall segmentation quality as the number of frames increases from 1 to 6. Overall, using a moderate number of frames, such as 6 or 8, achieves optimal VOS performance on the DAVIS 2017 dataset.

**Varying the LoRA rank** Table 7d demonstrates the effect of the LoRA rank on transfer performance. When the LoRA is 32, SINE achieves acceptable performance with few trainable parameters. Further increasing the rank to 64 only leads to a marginal improvement. Considering the trade-off between performance and the number of trainable parameters, we select rank 32 as the default setting.

## 5 Discussion and Conclusion

**Conclusion** In this work, we point out the task ambiguity in in-context segmentation for the first time and present SINE, which simultaneously predicts multiple task-specific masks to address this problem. Leveraging the efficient design, SINE can utilize few trainable parameters to exhibit strong segmentation abilities.We hope SINE can promote the development of in-context segmentation. **Limitation** and more **Discussions** are provided in Appendix A.

## Acknowledgement

This work is partially supported by the National Key R&D Program of China(NO.2022ZD0160101) and the National Natural Science Foundation of China (No. 62206244).

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

# Appendix

## A  Discussions and Limitation

**Comparison to SegGPT** Both SINE and SegGPT are in-context segmentation models. SINE's flexibility is comparable to SegGPT for various segmentation tasks. SINE can handle instance segmentation, which fails in SegGPT. SegGPT cannot deal with task ambiguity. The pixel output of SegGPT is not the final result and necessitates complex post-processing that converts the RGB output to mask. SINE avert this problem by predicting the mask directly. Stitching reference and target in SegGPT limits its capacity to process high-resolution images. SINE does not have this problem.

**Comparison to SAM** SINE and SAM offer different paths to the segmentation foundation model. SAM provides semantic-free promptable segmentation, while SINE handles semantic in-context segmentation. With different emphases, SINE and SAM can complement each other. Take auto-labeling as an example. SAM labels objects (*e.g.*, a dog) in the first image. SINE could use that image as an in-context example to label subsequent images to reduce costs.

**Limitation** As the first work to study ambiguity in in-context segmentation, SINE focuses on resolving ambiguities among ID, instance, and semantic segmentation tasks (as these are more important and commonly used). More complex ambiguities, such as full objects and parts, spatial position, category, and color, can be addressed by incorporating multimodal in-context examples (e.g., image and text). In addition, SINE has a performance gap compared with SegGPT in video segmentation. We think the reason behind this performance gap is that SegGPT trains all model parameters (300M), while SINE uses a simpler in-context fusion module and fewer learnable parameters (19M), limiting its ability to handle inter-frame relations in complex videos. Although SINE currently has limitations in learning complex inter-frame relationships in videos, we believe that by designing a more suitable In-Context Interaction module, the current paradigm holds greater potential for solving in-context segmentation tasks.

**Broader Impacts** Our Method is built upon open-source foundation models, significantly reducing carbon emissions. We do not foresee any obvious undesirable ethical or social impacts now.

## B  Implementation Details

**Training Details.**   We train our model with diverse segmentation datasets, including semantic, instance, and panoptic. The sampling weight for each dataset is 0.14 (COCO panoptic), 0.14 (ADE20K panoptic), 0.18 (COCO instance), and 0.54 (Objects365 instance). SINE uses two kinds of in-context pairs, selecting two images including the same category objects as the in-context pair or using two transformed views of the same image as the in-context pair. The probability is 0.5 for both in-context pairs.

Vision foundation models [18, 47, 44, 33] have demonstrated amazing visual representation capabilities, encouraging us to explore their performance in a wider range of applications. We deploy the frozen DINOv2 (ViT-L) [44] with 304M parameters as the Transformer encoder of SINE and train the In-Context Fusion module and the lightweight Transformer decoder with only 19M trainable parameters. The decoder contains six M-Former blocks. The model size of SINE is comparable with SegGPT (307M). SINE has fewer trainable parameters, leading to more efficient training. We randomly initialize the trainable parameters and train SINE about 50K steps with 64 batch sizes. We use Adam [36] optimizer and employ $\beta_1 = 0.9, \beta_2 = 0.999$ for optimization. We use a linear learning rate scheduler with a base learning rate of $1e-4$ and a warmup of 100 steps. The weight decay is set to 0.05. For data augmentation, we use random horizontal flipping and the large-scale jittering (LSJ) [13] augmentation with a random scale sampled from range 0.1 to 2.0 followed by a fixed size crop to $896 \times 896$. Our model is trained for 5 days by using 8 NVIDIA V100 GPUs.

**Evaluation.** The in-context examples are from the support samples for few-shot semantic segmentation, the training set for few-shot instance segmentation, and the first video frame for video object segmentation. The semantic prototypes can be seen as the classifier. The post-processing is similar to Mask2Former [5], but we use the predictions of the ID queries for VOS and the predictions of instance queries for instance segmentation, respectively.

# C  Additional Results

## C.1  Comparison of SINE and SegGPT without Objects365.

Table 8 compares the one-shot semantic segmentation results of training SINE using only ADE20K and COCO with SegGPT. SINE outperforms SegGPT on three benchmarks. Notably, SINE achieves 10% higher mIoU than SegGPT on LVIS-92i, indicating stronger class generalization capability in real-world image segmentation compared to SegGPT.

| Methods | COCO-20$^i$ | PASCAL-5$^i$ | LVIS-92$^i$ |
|---------|-------------|--------------|-------------|
| SegGPT  | 56.1        | 83.2         | 18.6        |
| SINE    | 67.1        | 86.3         | 28.8        |

**Table 8** – Comparison of SINE and SegGPT without Objects365.

## C.2  Impact of Different Backbones.

We select DINOv2-S, DINOv2-B, DINOv2-L, and CLIP-L [47] to explore the impact of different backbones. The conclusions are as follows: 1) DINOv2 is better than CLIP. DINOv2 achieves better performance than CLIP because it has general matching capabilities at both image and patch levels, allowing it to better understand complex contextual information between images. In contrast, CLIP captures image-text similarity, making it difficult to capture relationships between images, leading to poorer performance. 2) Larger DINOv2 model performs better: larger DINOv2 models have stronger representation capabilities, making it easier to capture contextual relationships, thus improving performance. This also indicates that SINE is scalable with the enhanced capabilities of the encoder.

| Methods | Backbone | COCO-20$^i$ | PASCAL-5$^i$ | LVIS-92$^i$ |
|---------|----------|-------------|--------------|-------------|
| SegGPT  | -        | 56.1        | 83.2         | 18.6        |
| SINE    | DINOv2-S | 56.8        | 81.4         | 26.7        |
|         | DINOv2-B | 61.7        | 84.1         | 29.5        |
|         | DINOv2-L | 64.5        | 85.4         | 31.2        |
|         | CLIP-L   | 34.8        | 57.3         | 16.1        |

**Table 9** – Impact of Different Backbones.

## C.3  Generalization of SINE on One-shot Part Segmentation.

As shown in Fig. 5, SINE can perform part segmentation like SegGPT when including PACO as training data. We evaluate SINE for one-shot part segmentation on PASCAL-Part [42] following Matcher [34] for the data pre-processing and evaluation. Compared with SegGPT [56], SINE achieves competitive performance.

| Methods | animals | indoor | person | vehicles | mean |
|---------|---------|--------|--------|----------|------|
| SegGPT  | 22.8    | 50.9   | 31.3   | 38.0     | 35.8 |
| SINE    | 22.7    | 61.8   | 21.7   | 38.5     | 36.2 |

**Table 10** – Results of one-shot part segmentation on PASCAL-Part.

## C.4  Generalizability of SINE beyond Semantically Similar Objects.

Fig. 6 shows SINE's capability in handling complex interaction relationships. In Fig. 6(a), the reference consists of multiple images, each containing different objects (box, cup, keyboard, mouse). When using these as in-context examples, SINE can segment one or more semantically different objects on a desk. In Fig. 6(b), with a reference containing only one object, the in-context example cannot represent complex interactions, and thus no segmentation result is provided. In Fig. 6(c), replacing multiple single-object images with a single image containing multiple objects yields the same effective results.

## C.5 Visualizations

Fig. 7 shows further visual comparisons. For video tasks, SINE reduces tracking failures from intersections, viewpoint changes, and occlusions. SINE addresses task ambiguity, preventing errors in semantic segmentation where SegGPT fails, as shown in the second set of comparison results in Fig. 7(a). In real-world image segmentation, SINE exhibits better class generalization than SegGPT, matching the LVIS-92i comparison in Table 1. SINE can effectively alleviate the prompt ambiguity and produce more accurate segmentation results. We provide more visualizations for example-based semantic segmentation in Fig. 8, example-based instance segmentation in Fig. 9, video object segmentation in Fig. 10, semantic segmentation on ADE20K, and panoptic segmentation on COCO in Figure 11.

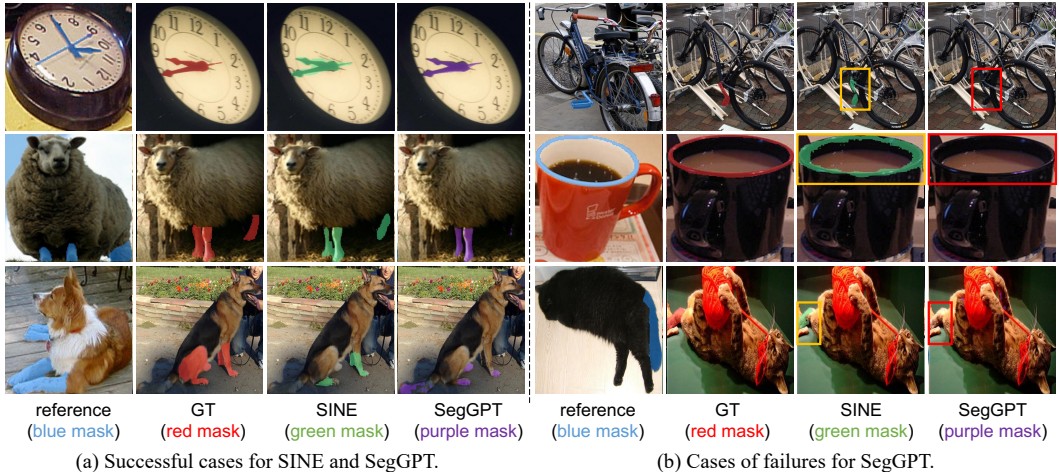

| reference (blue mask) | GT (red mask) | SINE (green mask) | SegGPT (purple mask) | reference (blue mask) | GT (red mask) | SINE (green mask) | SegGPT (purple mask) |

(a) Successful cases for SINE and SegGPT.   (b) Cases of failures for SegGPT.

**Figure 5** – Visualization of part segmentation.

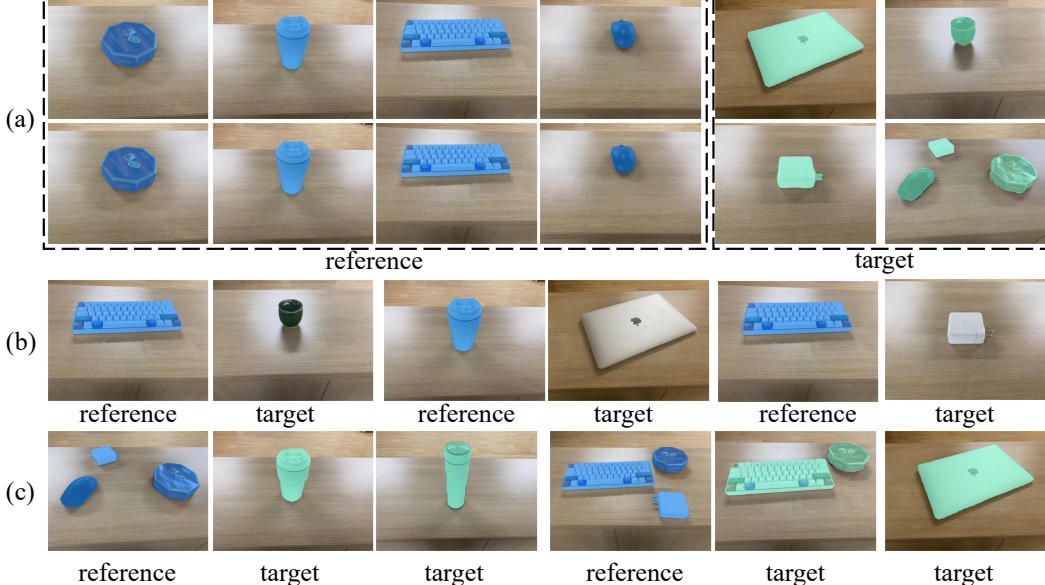

**Figure 6** – Generalizability of SINE beyond semantically similar objects. (a) and (c) SINE can recognize and segment another object when given successive images (or one image) including different objects. (b) SINE cannot output masks when given one reference image with a different object.

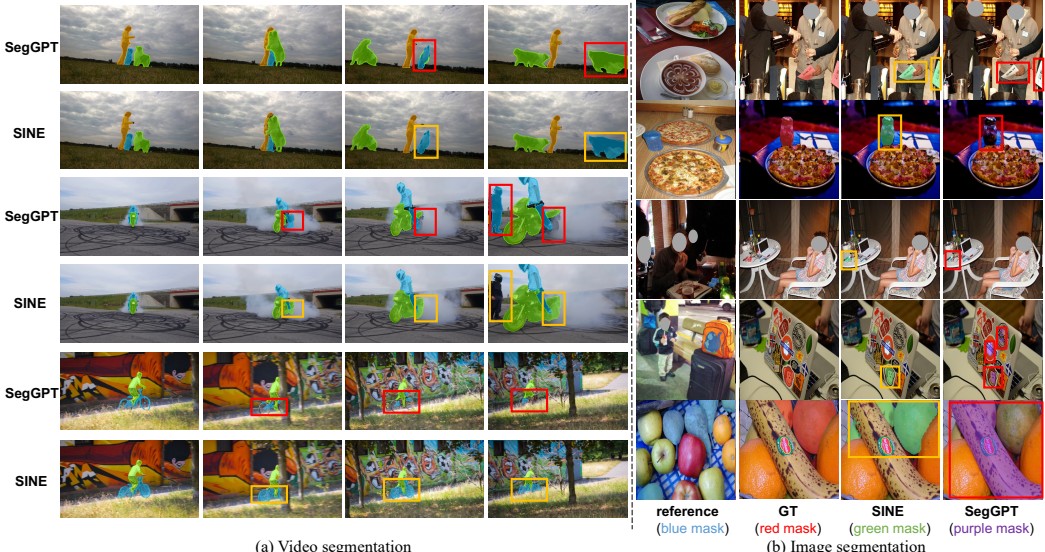

(a) Video segmentation

(b) Image segmentation

**reference** (blue mask) **GT** (red mask) **SINE** (green mask) **SegGPT** (purple mask)

**Figure 7** – Visualization comparisons between SINE and SegGPT on video and image tasks. For video task, SINE can effectively alleviate the tracking failure issues caused by crossing, perspective changes, occlusion, etc. In real-world image segmentation, SINE has a stronger category generalization capability compared to SegGPT and can effectively alleviate the ambiguity problem.

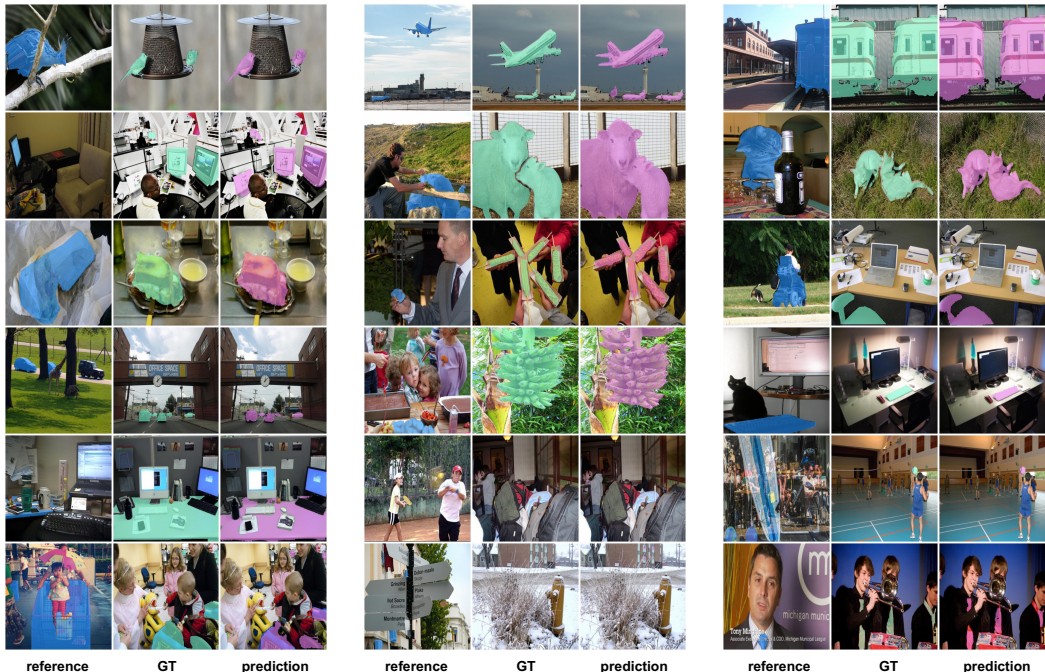

**reference** **GT** **prediction** **reference** **GT** **prediction** **reference** **GT** **prediction**

**Figure 8** – Visualization of example-based semantic segmentation.

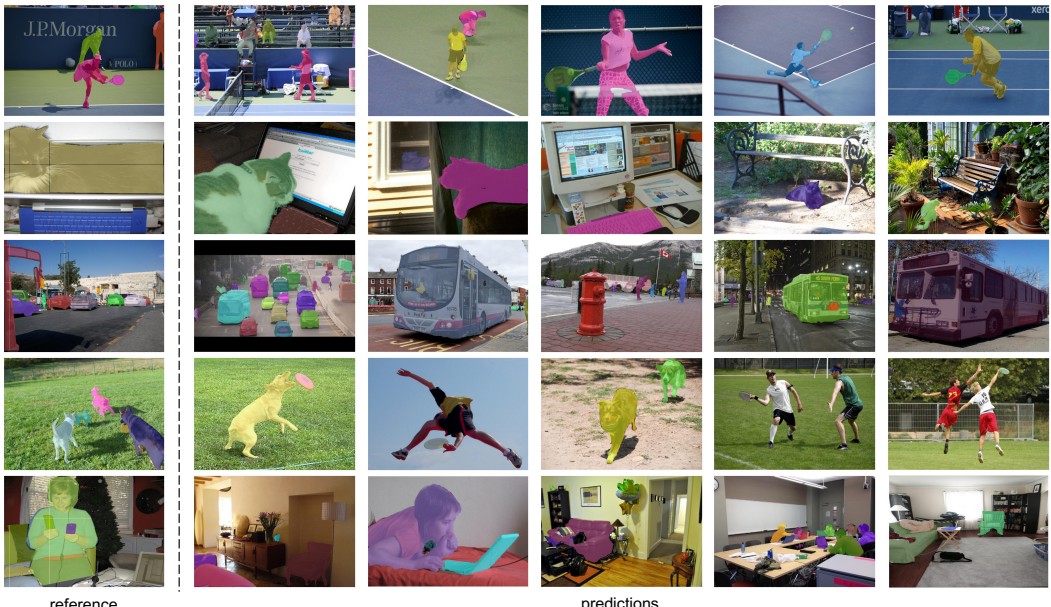

reference                                                                    predictions

**Figure 9** – Visualization of example-based instance segmentation.

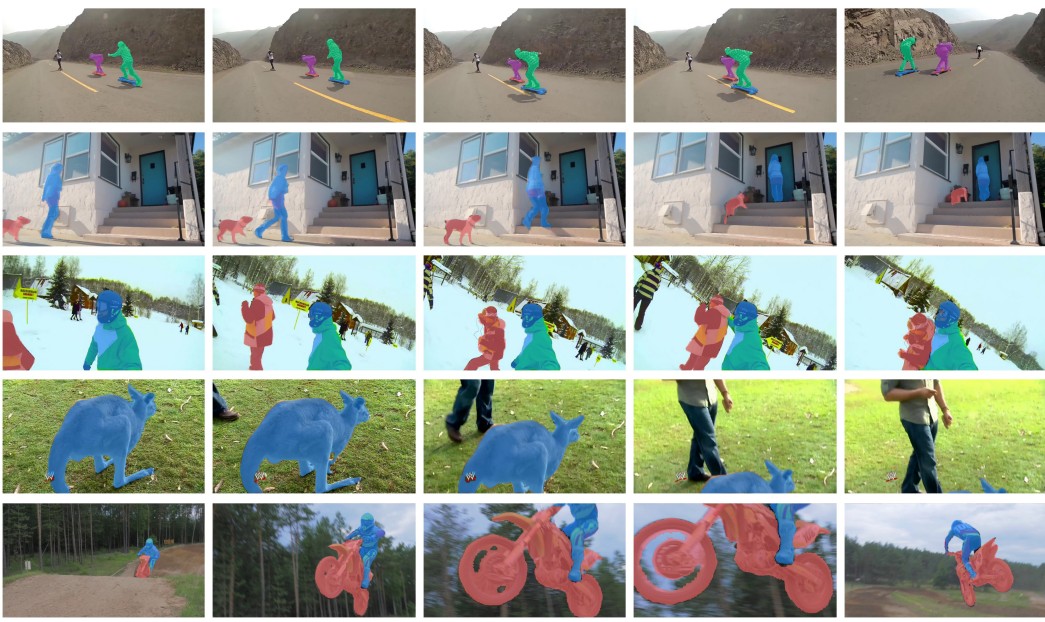

**Figure 10** – Visualization of video object segmentation.

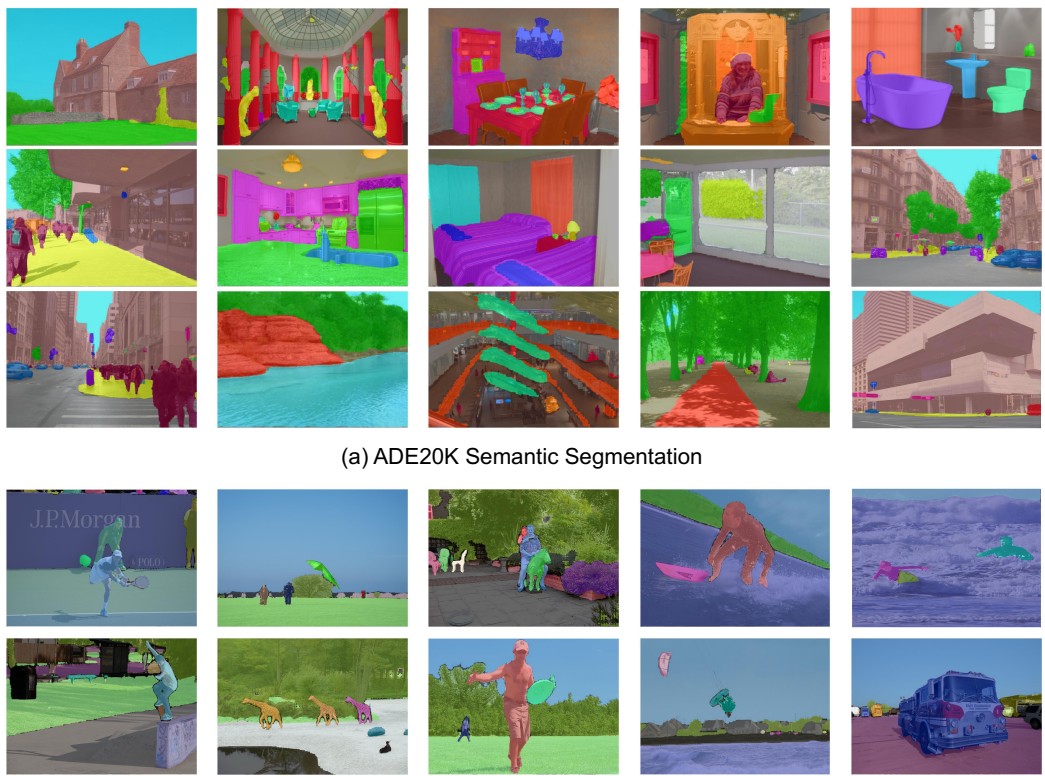

(a) ADE20K Semantic Segmentation

(b) COCO Panoptic Segmentation

**Figure 11** – Visualization of semantic segmentation on ADE20K, and panoptic segmentation on COCO.

