# OpenReview forum: "A Simple Image Segmentation Framework via In-Context Examples"
_NeurIPS.cc/2024/Conference — NeurIPS 2024 poster_

### Official Review · Reviewer_2YmX · 2024-07-07

**Soundness:** 2
**Presentation:** 2
**Contribution:** 2
**Rating:** 3
**Confidence:** 5

**Summary:**

This paper proposes a generalist segmentation model, dubbed as SINE, for a variety of segmentation tasks.

The general idea is to harness the in-context examples, and to alleviate the task ambiguity.

Specifically, an in-context interaction module, a matching Transformer and a Hungarian algorithm is devised to realize the above objectives.

Extensive experiments on a variety of datasets and segmentation tasks show its effectiveness.

**Strengths:**

+ This paper is overall well-written and easy-to-follow.

+ The proposed method and module design is rationale and effective.

+ The experiments and validation are extensive.

**Weaknesses:**

- The motivation does not match the methodology design properly. Specifically, the authors claim that the proposed method focuses on alleviating the task ambiguity. Unfortunately, throughout the module design, the in-context fusion has little relevance to the task-level ambiguity. Besides, the two losses in the M-Former are still implemented on the object level.

- The proposed method clearly lacks theoretical insight on how the task ambiguity is modeled and handled in theory. The overall module designs are common in visual representation learning, which is less relevant to the task-specific guidance or the task-level ambiguity.

- The Transformer design along with the Hungarian algorithm is not uncommon in modern Transformer based pipeline design such as DERT.

- From the performance perspective, compared with the prior in-context segmentation methods, the proposed SINE does not show a clear improvement in many cases, for example:

(1) Table 1, few-shot segmentation, on three out of six experiments, SINE is weaker than either Painter, SegGPT or recent ICLR works.

(2) Table 4, video object segmentation, the propose method is clearly inferior to the state-of-the-art by a large margin on two out of three datasets.

- The ablation study Table 5c is confusing, and more details or experiments need to be clarified. For example:

(1) In-context fusion is only a part of in-context interaction module, while in Table 5c the authors seem to only include the impact of fusion part.

(2) The specific components in M-Former also need further study, such as the specific loss functions and the phototype length and etc.

- Eq.3 Hungarian loss. Do other types of commonly-used loss functions also achieve similar performance? More discussion and comparison is needed.

- The visual results are not sufficient enough. Although the authors provide a lot of segmentation predictions in the supplementary material, only the results from the proposed method along with ground truth are provided. Please provide and compare the visual results from other state-of-the-art methods.

**Questions:**

- Q1：The motivation does not match the methodology design properly. The in-context fusion has little relevance to the task-level ambiguity.

- Q2: The two losses in the M-Former are still implemented on the object level.

- Q3: The proposed method clearly lacks theoretical insight on how the task ambiguity is modeled and handled in theory. The overall module designs are common in visual representation learning, which is less relevant to the task-specific guidance or the task-level ambiguity.

- Q4: The Transformer design along with the Hungarian algorithm is not uncommon in modern Transformer based pipeline design such as DERT.

- Q5: Limited performance: Table 1, few-shot segmentation, on three out of six experiments, SINE is weaker than either Painter, SegGPT or recent ICLR works.

- Q6: Limited performance:

- Q7: Table 4, video object segmentation, the propose method is clearly inferior to the state-of-the-art by a large margin on two out of three datasets.

- Q8: Clarify ablation study in Table 5c. (1) In-context fusion is only a part of in-context interaction module, while in Table 5c the authors seem to only include the impact of fusion part.
(2) The specific components in M-Former also need further study, such as the specific loss functions and the phototype length and etc.

- Q9: Eq.3 Hungarian loss. Do other types of commonly-used loss functions also achieve similar performance? More discussion and comparison is needed.

- Q10: The visual results are not sufficient enough. Please provide and compare the visual results from other state-of-the-art methods.

**Limitations:**

Though providing a discussion, the reviewer believes that the discussion is not proper.

Some potential negative impacts such as job displacement brought by generalist model should be discussed.

---

> ### Author Rebuttal · Authors · 2024-08-05
>
> >W1,Q1,Q2: Motivation does not match methodology design properly.
>
> The motivation and methodology design are related and clear (supported by **sZmi,rhP3**).
>
> - The goal of In-Context Fusion is to establish the correlations between reference and target (see Line 163-164), understanding the complex information within the context, which is crucial for the in-context segmentation task (supported by **sZmi**).
> - The proposed M-Former is designed to address task ambiguity in prompts, with the challenges for M-Former regarding task ambiguity analyzed in Lines 181-185.
> - This paper discusses the relationships between different tasks in detail (Line 112-117) and unifies them using instance segmentation, allowing different tasks to use the same model. Therefore, the loss is reasonable (supported by **nRMP**).
>
> The paper clearly states the motivation of all designs, and their effectiveness is verified through experiments. This has also been recognized by other reviewers, aligning our method's design with our motivation.
>
> >W2,Q3: Insights and relevance.
>
> The contributions and insights are as follows (please see **General Response** for details):
> 1. Ambiguities in Visual Prompting
> 2. Investigating the Capabilities of VFMs
> 3. Lightweight and Effective Decoder
>
> **We believe the module designs are relevant to our motivation.** We design M-Former to address task ambiguity. Information from different tasks can interfere, causing incorrect predictions (Lines 181-185).
> *Mask2Former and DETR are not designed for in-context segmentation, making it difficult to address these challenges.*
> M-Former's dual-path structure and shared SA enable effective information interaction and prevent task interference, balancing efficient in-context segmentation decoding and resolving task ambiguity in prompts. In Table 5c, compared to Mask2Former, M-Former effectively resolves task ambiguity, resulting in improvements across various tasks.
>
> >W3,W6,Q4,Q9: Transformer along with Hungarian algorithm is not uncommon in modern Transformer based pipeline. Eq.3 Hungarian loss.
>
> Hungarian Loss: **We do not claim the loss as a contribution.** Please refer to the General Response or Lines 65-71 of the paper for our contributions. The Hungarian algorithm has been widely used[A,B,C], and our use of it aligns with these works, making it reasonable and **not a weakness**. The loss with one-to-many matching used by the traditional method[D,E] will introduce NMS, which is not suitable for Transformer based method.
>
> Transformer design: We analyze task ambiguity challenges (Lines181-185) and design M-Former to address them. The dual-path design and Shared SA ensure efficient decoding and prevent information confusion across tasks of different granularity. Unlike previous methods, SINE improves performance with fewer parameters, demonstrating our method's novelty. See our **response to Reviewer rhP3 for more details**.
>
> >W4(1),Q5,Q6: SINE is weaker than other works on FSS.
>
> Our experiments show that SINE achieves significant improvements across multiple tasks and benchmarks (supported by **sZmi and nRMP**) and demonstrates solid practical utility (supported by **rhP3**).
>
> The table below shows the **average performance** of Table 1, SINE achieves the best performance. Therefore, SINE's improvements are significant.
>
> ||one-shot|few-shot|
> |---|---|---|
> |Painter|35.9|36.0|
> |SegGPT|52.6|61.0|
> |Matcher|42.9|50.4|
> |SINE|60.4|62.6|
>
> >W4(2),Q7: VOS.
>
> See our **response to sZmi‘s W3**.
>
> >W5(1),Q8: Impact of fusion part in Table 5c.
>
> Mask Pooling part does not contain any parameters. Learning in-context information is demonstrated in In-Context Fusion (see Lines 170-171). Mask Pooling is a necessary operation for SINE and cannot be removed. Hence, the corresponding ablation was not performed in Table 5.
>
> >W5(2),Q8: Studies of specific loss, phototype length.
>
> Phototype: The number of prototypes is determined by the prompts (reference images and masks) and is not a settable hyperparameter. The number equals the total count of different semantic categories in the in-context examples.
>
> Loss: Hungarian loss has been widely validated in object detection and segmentation[A,B,C]. We use this loss to align with Mask2Former and facilitate comparison (see Table 5(c)). Loss with one-to-many matching[D,E] is not suitable. Other hyperparameters, e.g., query dimension, are also aligned with Mask2Former to eliminate their impact.
>
> >W7,Q10: More visualization comparison results.
>
> Fig.3 in attached PDF shows further visual comparisons. For video tasks, SINE reduces tracking failures from intersections, viewpoint changes, and occlusions. SINE addresses task ambiguity, preventing errors in semantic segmentation where SegGPT fails, as shown in the second set of comparison results in Fig.3(a). In real-world image segmentation, SINE exhibits better class generalization than SegGPT, matching the LVIS-92i comparison in Table 1.
> ___
>
> >Limitation 1:The reviewer believes that the discussion is not proper.
>
> **We believe the discussion is proper.** We discuss the connections and differences between SINE and related works (SegGPT and SAM), and the limitations of SINE(Line478-494). We believe these discussions are necessary. **If the reviewer finds any inaccuracies**, please specify them in detail so we can further address them.
>
> >Limitation 2: Some potential negative impacts such as job displacement brought by generalist model should be discussed.
>
> We do not foresee any obvious undesirable ethical or social impacts now. We believe that generalist models are powerful tools for efficient production and will create more job opportunities rather than causing job displacement.
>
> [A] End-to-end object detection with transformers. ECCV2020.
>
> [B] Per-Pixel Classification is Not All You Need for Semantic Segmentation. NIPS2021.
>
> [C] Masked-attention mask transformer for universal image segmentation. CVPR2022.
>
> [D] Mask r-cnn. ICCV2017.
>
> [E] SOLO: Segmenting Objects by Locations. ECCV2020.

---

> ### Comment · Reviewer_2YmX · 2024-08-07
> **The rebuttal does not address my substantial concerns**
>
> Thanks the authors for the rebuttal.
>
> Unfortunately, the rebuttal keeps repeating the comments and views from other reviewers, instead of seriously addressing the specific concerns raised by me. For example:
>
> - **Q1**：I am not convinced from the text from the submission mentioned by the authors. The text still considers the object level learning. No rigid defintion or formulation is given, especially to the level of task. Neither does the substantial relevance to the task-level ambiguity.
>
> - **Q2**: *The two losses in the M-Former are still implemented on the object level.* How do them relate to the task-level and the ambiguity? The rebuttal does not directly answer this question. Instead, it just keeps highlighting the view from other reviewers and mentions some void and lackluster general contributions.
>
> - Besides, the authors in the rebuttal claim *Lightweight and Effective Decoder*. But the evidence in the main text supporting this aspect is not enough. The parameter comparison is only an ablation study, not with other types of decoders.
>
> - **Q3**: The authors keep repeating the application value and insight in the rebuttal. However, how is the task ambiguity modeled, formulated and defined, and whether some theorical support can be found, are neither clarified.
> Besides, the concern *The overall module designs are common in visual representation learning, which is less relevant to the task-specific guidance or the task-level ambiguity* is unaddressed.
>
> - **Q4**: *We do not claim the loss as a contribution* does not necessarily mean *the devised representation learning pipeline is techniqually novel*. The authors do not address ths aspect directly in the rebuttal. For example, would the authors acknowledge a paper's novelty and significance, if it claims to use U-Net with minor modifications for another task / application such as industrial segmentation for the first time?
>
> - **Q5 \& Q7**: I acknowledged that in some tables the proposed SINE shows state-of-the-art performance. However, my question also raises. *Table 1, few-shot segmentation, on three out of six experiments, SINE is weaker than either Painter, SegGPT or recent ICLR works.* The authors do not address this in the point-by-point response.
>
> - I acknowledge from the rebuttal that the average performance is the best, but this does not contradict to the fact that, it shows significantly inferior performance in many of my listed settings. In fact, it further raises the concerns on whether it is stable or generalized enough.
>
> - **Q8 \& Q9**: *Using the same loss as other works for fair evaluation* is not a proper excuse to not study the impact of some common loss types. This aspect is unaddressed.
>
> - **Q10**: Only limited visual results on limited datasets from SegGPT is provided. My concerns on more visual results from state-of-the-art methods are not addressed.
>
> - Minor issue: **Limitation**: Efficient production definately leads to the loss of some old, tradtional and off-the-shelf jobs. This does not contradict *create more job opportunities*. Why is it difficult for the authors to acknowledge this aspect?
>
> To conclude, my substantial concerns are not properly addressed. I keep my original rating and recommend clear reject this paper.

---

> ### Author Response · Authors · 2024-08-08
> **More detailed explanation**
>
> Thank you for your comments. Due to the rebuttal's character limit, we couldn't fully address your concerns. Below, we provide a detailed explanation.
>
> >No rigid defintion or formulation is given, especially to the level of task. Neither does the substantial relevance to the task-level ambiguity.
>
> The formulation of SegGPT can be represented as:
>
> $f(x_r, y_r, x_t) \rightarrow y_t,~~~   y_t \in \{task_1, task_2, task_3\}$
>
> When the given prompt $(x_r, y_r)$ cannot precisely indicate a specific task, the prompt is ambiguous, as shown in Fig. 1 of the paper. When SegGPT performs a task (e.g., $task_1$) with an ambiguous prompt, it might incorrectly output the results of $task_2$ or $task_3$. **This is an important and unexplored problem.**
>
> The formulation of SINE is as follows::
>
> $f(x_r, y_r, x_t) \rightarrow \{ y_t^{task_1}, y_t^{task_2}, y_t^{task_3} \}$
>
> By providing results for all tasks, SINE avoids incorrect predictions caused by prompt ambiguity. Fig. 4(a) in the paper presents a visual comparison.
>
> >The text still considers the object level learning. The two losses in the M-Former are still implemented on the object level.
>
> We address the ambiguity between ID, instance, and semantic segmentation. These tasks can be converted into instance segmentation. Using instance/object level segmentation allows for unified loss forms across different tasks. M-Former introduces a dual-path decoder and shared self-attention with a mask (Fig. 2, top right), enabling effective information interaction and preventing task interference. This balances in-context segmentation decoding and resolves prompt ambiguity.
>
> Our design unifies training for different tasks and prevents task information contamination during decoding.  During inference, the trained network can perform different tasks, which we believe is an advantage of our method.
>
> >Lightweight and Effective Decoder
>
> Compared to SegGPT's 300M training parameters, our model has only 19M. Without the encoder, Mask2Former has 21M parameters. Thus, our method is more lightweight.
>
> > We do not claim the loss as a contribution does not necessarily mean the devised representation learning pipeline is techniqually novel. Would the authors acknowledge a paper's novelty and significance, if it claims to use U-Net with minor modifications for another task / application such as industrial segmentation for the first time?
>
>
> We believe **the value of a paper lies in whether it provides academic insights to the research community.** For example, Marigold (CVPR 2024, Best Paper Award Candidate) first demonstrated stable diffusion's significant generalization in depth estimation. However, Marigold made no changes to U-Net's structure or training strategy. This does not change our view that Marigold is an outstanding and inspiring work.
>
> In addition, our core claim is that we are the first to identify and explore task ambiguity in prompts, rather than making minor modifications for in-context segmentation. **We believe this will inspire other work.** The General Response summarizes our contributions and insights.
>
> > Table 1, few-shot segmentation, on three out of six experiments, SINE is weaker than either Painter, SegGPT or recent ICLR works.
>
> 1. SINE outperforms Painter on all datasets.
> 2. SegGPT is trained on COCO and PASCAL and uses a Context Ensemble strategy for few-shot settings. SINE is not trained on PASCAL and is not designed for few-shot settings. However, SINE achieves the best performance on 1-shot in COCO-20i and PASCAL-5i, and is comparable to SegGPT in few-shot settings.
> 3. On LVIS-92i, SegGPT is only weaker than Matcher because Matcher uses SAM, and its larger training set better aligns with the LVIS.
>
> > Eq.3 Hungarian loss. Using the same loss as other works for fair evaluation is not a proper excuse to not study the impact of some common loss types. This aspect is unaddressed.
>
> In the rebuttal, we explained that Hungarian loss is essential for Detection/Segmentation Transformer methods. These methods model detection/segmentation task as a set prediction problem, introducing one-to-one Hungarian matching. Traditional methods use a one-to-many matching for loss, which requires NMS and is not applicable here. Could the reviewer specify which loss should be used for comparison?
>
> > Limited visual results.
>
> The visual results in the rebuttal are limited to the **one-page** PDF requirement. We will provide more visual comparisons in the paper.
>
> > Limitation: Efficient production definately leads to the loss of some old, tradtional and off-the-shelf jobs. This does not contradict create more job opportunities. Why is it difficult for the authors to acknowledge this aspect?
>
> We will add this discussion in the paper. However, almost all AI model developments may cause such issues in the short term, but we cannot stop the advancement of AI because of this. We believe that, in the long run, the benefits of developing general models for human society outweigh the drawbacks.

---

> > ### Comment · Reviewer_2YmX · 2024-08-09
> > **Re: More detailed explanation**
> >
> > First of all, I would appreciate the authors' effort, so that we could start a constructive discussion on the weakness and its improvement.
> >
> > Regarding the specific responses, I've spent some time on it, and conclude that there are still multiple issues need to be clarified / resolved before raising up to the accept threshold.
> >
> > **No rigid defintion or formulation (part of continued Q1)**:
> >
> > - The SegGPT formulation is fine, as it is still an one-to-one mapping. However, is it really true that SINE is a one-to-three mapping? From the reviewer's understanding, in the experimental table, you still do the inference on one task/ dataset one time, right? If so, this is not a one-to-three mapping, and the correctness remains doubted.
> >
> > - From my view, the proposed SINE can implictly learn a rank between different types of tasks. Could the authors re-consider the formulation, and try to model the score of each task? In this way, it will improve the clarity and also make sense to the distingushment between different tasks.
> >
> > - On top of this, perhaps the authors can find a way to model the ambiguity between different tasks, which can in turn help explain the proposed SINE from a more theortical perspective.
> >
> > I would expect if some aspects can be made on the above points, so as to improve both the clarity and the theotical insight.
> >
> > **Object level learning (Q2)**: There are still some remaining issues before resolving this question:
> >
> > - In my view, *instance* in instance segmentation is typically object-level. Could the authors maybe justify more on *Using instance/object level segmentation allows for unified loss forms across different tasks*?
> >
> > - *M-Former introduces a dual-path decoder and shared self-attention with a mask (Fig. 2, top right), enabling effective information interaction and preventing task interference*. This explanation does not convince me. The interaction is still between instance representations, right? It just exploits the long-rang dependencies between different instances. How could it benefit the task level?
> >
> > **Lightweight and Effective Decoder**: This concern has been well addressed, after having a comparison with existing paradigm.
> >
> > **Representation learning novelty (Q3\&Q4)**: There is so far no strong argument to address this perspective. As the representation learning pipeline is very ordinary and common, and no theoretical insight can be demonstrated so far, I still believe this work is typically beyond the standard of top-tier conference.
> >
> > **Limited performance (Part of Q5\&Q7)**: The explanation does not alter the fact of limited performance on three out of six experiments. Maybe the authors have to make significant revision on this aspect in the main text, to discuss what is the reason.
> > Besides, if there is some way to make a fair evaluation between them?
> >
> > **Regarding the loss design**: Can some other common mechanisms such as IoU based Non-Maximum Suppression, Greedy Algorithm, Positive Sample Mining be inspected in the loss design? Anyway, this is a minor issue, but I insist that the loss and overall idea is ordinary.
> >
> > **Limited Visual Results**: Hope the authors can further address this part later, as it is still a minor weakness of this work.
> >
> > **Limitation Discussion**: Thanks the authors for acknowledging this aspect and making the adapations accordingly.

---

> > > ### Author Response · Authors · 2024-08-10
> > >
> > > Thank you very much for your efforts and time in further discussing our work. Below, we provide a more detailed clarification regarding the remaining concerns.
> > >
> > > >Q1 No rigid defintion or formulation (part of continued Q1):
> > >
> > > Our goal is to address the ambiguity problem by using the following formulation, which outputs the results for all tasks simultaneously:
> > >
> > > $f(x_r, y_r, x_t) \rightarrow \{ y_t^{ID}, y_t^{Ins}, y_t^{Sem} \}$
> > >
> > > As the reviewer mentioned, since SINE’s learning process operates at the object level (including ID and instance), the formulation for SINE is:
> > >
> > > $f(x_r, y_r, x_t) \rightarrow y_t \rightarrow \{y_t^{ID}, y_t^{Ins}\}$
> > >
> > > The losses for instance and ID segmentation are clear and well-defined, as shown in Equations (3) and (4) of the paper. Therefore, the above formulation holds. We believe the primary concern here lies with semantic segmentation.
> > >
> > > For semantic segmentation, SINE does not directly output $y_t^{Sem}$. Instead, during inference, we can obtain it by merging the instance segmentation predictions $y_t^{Ins}$. Specifically,
> > >
> > > $y_t^{Ins} = \{ P_{mask}^{Ins} \in R^{S \times  H \times W}, P_{class}^{Ins} \in R^{S \times M} \}$
> > >
> > > $P_{mask}^{Ins}$ and $P_{class}^{Ins}$ represent the mask and class predictions for instance segmentation. Here, $S$ is the number of instance queries, and $M$ is the number of semantic prototypes, i.e., the number of candidate categories.
> > >
> > > The semantic segmentation result $y_t^{Sem} \in  R^{M \times  H \times W}$ can be concisely expressed using the following matrix multiplication formula:
> > >
> > > $y_t^{Sem} = P_{class}^T \times P_{mask}^{Ins} $
> > >
> > > Where $P_{class}^T \in R^{M \times S}$ is the transpose of $P_{class}^{Ins} $, $ y_t^{Sem} \in R^{M \times H \times W}$ is the semantic segmentation result before applying argmax.
> > >
> > > This formula indicates that the segmentation map for each category $m$, denoted as $y_t^{Sem}[m]$, is obtained by a weighted sum of all instance masks $P_{mask}[s]$ with the corresponding class probabilities $P_{class}[s, m]$.
> > >
> > > Therefore, the SINE formulation can be expressed as:
> > >
> > > $f(x_r, y_r, x_t) \rightarrow \{ y_t^{ID}, y_t^{Ins}, y_t^{Sem} \}$
> > >
> > > **Through the above derivation, we hope to provide a clearer answer to the reviewer's question.**
> > >
> > > >Is it really true that SINE is a one-to-three mapping?
> > >
> > > Yes, during the inference stage, SINE can provide $ \{y_t^{ID}, y_t^{Ins},y_t^{Sem} \}$ for any input $(x_r, y_r, x_t)$.
> > >
> > > >In the experimental table, you still do the inference on one task/dataset one time, right?
> > >
> > >  SINE performs inference on any dataset by predicting all three tasks simultaneously. We select the relevant output based on the specific task. For instance, Table 1 uses semantic segmentation $y_t^{Sem}$ ​, Tables 2 and 3 use instance segmentation  $y_t^{Ins}$, and Table 4 uses Object ID $y_t^{ID}$ on the VOS dataset. Thank you for pointing this this. We will clarify it further in the revised version.
> > >
> > >  >From my view, the proposed SINE can implictly learn a rank between different types of tasks.
> > >
> > >  The above derivation shows that SINE provides results for all tasks simultaneously, rather than learning a rank between different types of tasks.
> > >
> > > >On top of this, perhaps the authors can find a way to model the ambiguity between different tasks, which can in turn help explain the proposed SINE from a more theortical perspective.
> > >
> > > Thank you for the helpful suggestions in formulating SINE. We believe this makes SINE clearer and more interpretable from a theoretical perspective.
> > >
> > > >Object level learning (Q2): There are still some remaining issues before resolving this question: In my view, instance in instance segmentation is typically object-level. Could the authors maybe justify more on Using instance/object level segmentation allows for unified loss forms across different tasks?
> > >
> > > Beyond the previous clarification of SINE's formulation, we would like to further address the reviewer's concerns.
> > >
> > > SINE targets three tasks: ID segmentation, instance segmentation, and semantic segmentation. During training, we focus on learning instance and ID segmentation, as defined by Equations (3) and (4). The differences in losses lies only in the matching strategy between predictions $y_t$ and ground truth, while the loss functions remain the same, making the form of the two losses unified. Semantic segmentation is derived from instance segmentation results by combining masks of instances within the same category.

---

> > > > ### Author Response · Authors · 2024-08-10
> > > >
> > > > >M-Former introduces a dual-path decoder and shared self-attention with a mask (Fig. 2, top right), enabling effective information interaction and preventing task interference. This explanation does not convince me...
> > > >
> > > > In M-Former, interactions may occur between ID/instance level queries, semantic level prototypes, and image features. M-Former avoids the contamination of information from different granularities, thereby benefiting task performance. Specifically:
> > > >
> > > > - Dual-Path Structure: M-Former's input includes both queries and prototypes. The queries are responsible for instance-level predictions, while the prototypes, derived from the reference image, need to maintain the semantic-level category information from the reference. Since their roles differ, SINE separates the processing of queries and prototypes, forming a dual-path structure. The query branch includes SA, CA, and FFN, ensuring effective information interaction between the queries and image features, allowing the model to learn object-level information. This process avoids interference from the semantic-level information contained in the prototypes.
> > > >
> > > > - Shared SA: To keep the model lightweight, M-Former shares the SA between queries and prototypes. The queries include both ID and instance queries. While both operate at the object level, they represent different levels of information granularity. For example, as shown in Fig. 1 of the paper, an ID query might represent Prof. Geoffrey Hinton, whereas an instance query could represent any object categorized as a "person." The prototype, meanwhile, represents the average information for the category "person." If the ID query for Hinton were to be confused with broader information, it might lead to incorrect predictions. Therefore, it’s crucial to prevent interaction between ID queries and the more coarse-grained instance queries or prototypes. A well-designed attention mask (Fig. 2, top right) effectively mitigates this issue.
> > > >
> > > > In summary, M-Former uses a dual-path decoder and shared SA to enable effective information interaction and preventing task interference. In contrast, DETR and Mask2Former's uniform approach to queries and prototypes of different granularities leads to information confusion and reduced performance.
> > > >
> > > > >Representation learning novelty (Q3&Q4): There is so far no strong argument to address this perspective.
> > > >
> > > > We acknowledge that SINE uses a Transformer decoder and Hungarian loss, methods employed by DETR, Mask2Former, and other recent approaches. However, unlike previous work mainly focused on model structure improvements, our paper addresses task ambiguity in in-context segmentation and how to effectively perform in-context segmentation with the DINOv2 representation. Utilizing Visual Foundation Models for various tasks is a growing trend. For instance, DINOv2+reg (ICLR 2024, Outstanding Paper Award) explores how DINOv2 effectively performs unsupervised object discovery.Additionally, M-Former was designed to better prevent task interference. Based on these contributions, we believe our work meets top-tier conference standards.
> > > >
> > > >
> > > > >Limited performance (Part of Q5&Q7): The explanation does not alter the fact of limited performance on three out of six experiments. Maybe the authors have to make significant revision on this aspect...
> > > >
> > > > SegGPT indeed achieved better few-shot performance on COCO-20i and PASCAL-5i due to its use of these datasets and the Context Ensemble strategy. However, SINE demonstrates superior class generalization. Evaluating both methods on LVIS-92i, which has 920 categories, shows that SINE outperforms SegGPT by about 10% in both one-shot and few-shot settings. This likely reflects SegGPT's overfitting to the 80 and 20 categories of COCO and PASCAL.
> > > >
> > > > On LVIS-92i, SINE only lags behind Matcher, which uses SAM-H, a model trained with more data (SA1B including 1B masks) and more parameters (945M), making a direct comparison somewhat unfair. For a fairer comparison, we replaced SAM-H with SAM-B using Matcher's public code. The table below shows that SINE performs better with fewer parameters on LVIS-92i fold0.
> > > >
> > > > ||params|LVIS-92i|
> > > > |---|---|---|
> > > > |Matcher SAM-H|945M|31.4|
> > > > |Matcher SAM-B|398M|28.1|
> > > > |SINE|317M|28.3|
> > > >
> > > > We will add more discussion in the revised version.
> > > >
> > > >
> > > > >Regarding the loss design: Can some other common mechanisms such as IoU based Non-Maximum Suppression...
> > > >
> > > > The Hungarian loss ensures each query matches exactly one ground truth, eliminating the need for NMS. While SINE could theoretically achieve one-to-many matching, allowing each ground truth to match multiple queries. However, we don't have enough time for this experiment during the discussion phase. That's really not the point of this paper either. In addition, addressing task ambiguity and interference through the loss function is a novel approach we will explore in future research.

---

> > > > > ### Comment · Reviewer_2YmX · 2024-08-12
> > > > > **Re: Official Comment by Authors**
> > > > >
> > > > > Thanks for your extensive effort to clarify the rest questions.
> > > > > However, there are still multiple questions need to be clarified before considering approching the borderline threshold.
> > > > >
> > > > > **No rigid defintion or formulation (part of continued Q1):** Most of the concerns have been well addressed. However, regarding the explanation *SINE performs inference on any dataset by predicting all three tasks simultaneously*.
> > > > >
> > > > > - Is there a mechanism to consider or to compute the confidence or the weight of each individual task?
> > > > >
> > > > > - From the reviewer's understanding on ambiguity and uncertainty, such amubiguity can be well resolved usually with some ways to measure the confidence of per-task. Otherwise the overall problem or the task is still deterministic, and the amubity may only exists verbally, downgradng the overall task formulation.
> > > > >
> > > > > - *can find a way to model the ambiguity between different tasks*. I am still confused on this aspect.
> > > > >
> > > > > **Object level learning (Q2)**: The first question has been addressed, and thanks for the authors. I appreciate the author's effort to provide extensive details on *Dual-Path Structure* and *shared SA*.
> > > > >
> > > > > - However, it is still *objective-level*, and does not connect to the relation to *task-level*.
> > > > >
> > > > > - Especially, some saying such as * enable effective information interaction and preventing task interference* is super volid, and lack clear explanation or explanation.
> > > > >
> > > > > **Representation learning novelty (Q3&Q4)**: Thanks for the clarification. However, as I mentioned earlier and also raised by another reviewer today, I still believe the novelties from both theoratical perspective and representation learning respective. This is particularly important, especially when the task formulation in Q1 and Q2 are still not well addressed.
> > > > >
> > > > > **Limited performance (Part of Q5&Q7):** I think at this stage these concerns have been well addressed. Please enrich the discussion accordingly.
> > > > >
> > > > > **Other Minor Issues such as losses and limitation**: Thanks for the authors to well address these issues. Yes, as I mentioned earlier, the discussion of different loss is a minor issues, which may be resolved in the future or even after the current form of the work.

---

> > > > > > ### Author Response · Authors · 2024-08-13
> > > > > >
> > > > > > Thank you very much for your efforts and time in further discussing our work. Below, we provide a more detailed clarification regarding the remaining concerns.
> > > > > >
> > > > > > ___
> > > > > >
> > > > > > >Q1 No rigid defintion or formulation (part of continued Q1): Is there a mechanism to consider or to compute the confidence or the weight of each individual task? From the reviewer's understanding on ambiguity and uncertainty, such amubiguity can be well resolved usually with some ways to measure the confidence of per-task. Otherwise the overall problem or the task is still deterministic, and the amubity may only exists verbally, downgradng the overall task formulation. can find a way to model the ambiguity between different tasks. I am still confused on this aspect.
> > > > > >
> > > > > > Different from uncertainty, to resolve the ambiguity requires an explicit specification of the prediction type (e.g. id/instance/semantic), instead of formulating the problem as a distribution estimation. Uncertainty estimation in segmentation task usually deals with pixel level empirical prediction risks induced either by the data or the model [A].
> > > > > >
> > > > > > Because of this inherent difference, our task ambiguity is not modelled in a similar way as the uncertainty.
> > > > > >
> > > > > > [A] What uncertainties do we need in bayesian deep learning for computer vision. NIPS2017.
> > > > > >
> > > > > >
> > > > > > >Object level learning (Q2): It is still objective-level, and does not connect to the relation to task-level. Especially, some saying such as * enable effective information interaction and preventing task interference* is super volid, and lack clear explanation or explanation.
> > > > > >
> > > > > > The input to M-Former includes ID queries ($q_{id}$), instance queries ($q_{ins}$), and the semantic prototype ($p_{sem}$), as shown in Fig. 2 of the paper. **$q_{id}$ and $q_{ins}$ are used for predicting ID segmentation and instance segmentation, respectively, and are task-specific.** To prevent information interference between $q_{id}$ and $q_{ins}$ / $p_{sem}$, we utilize shared SA with the well-designed attention mask to avoid their interaction.
> > > > > >
> > > > > > This object-level semantic segmentation approach significantly reduces fragmented masks, balances the training of masks of different sizes, and facilitates the unification of different tasks. Therefore, SINE also adopts an object-level approach for learning semantic segmentation.
> > > > > > By enabling interaction between instance-level $q_{ins}$ and semantic-level $p_{sem}$, we establish a stronger connection between instance segmentation and semantic segmentation.
> > > > > >
> > > > > > Thus, M-Former connects to the relation to task-level and effectively facilitates the interaction between $q_{ins}$ and $p_{sem}$ while preventing interference between $q_{id}$ and $q_{ins}$ / $p_{sem}$.
> > > > > >
> > > > > >
> > > > > > >Representation learning novelty (Q3&Q4): I still believe the novelties from both theoratical perspective and representation learning respective. This is particularly important, especially when the task formulation in Q1 and Q2 are still not well addressed.
> > > > > >
> > > > > > Different from uncertainty, which models probabilistic distribution, SINE explicitly presents all prediction results to address task ambiguity, thereby enhancing the practical usability of in-context segmentation: In widely used segmentation tasks such as semantic and instance segmentation, traditional methods require assigning a category label to the object, which becomes challenging when scaled to thousands of categories. *In-context segmentation overcomes these limitations by using visual concepts as annotations.* However, ambiguous prompts in in-context segmentation can lead to incorrect predictions, limiting its practical use. We are the first to consider and resolve task ambiguity in prompts for in-context segmentation, a significant contribution in both research and application, and we believe this challenge is inherently novel.
> > > > > >
> > > > > > In exploring the potential of DINOv2 for in-context segmentation, the challenge lies in how to maximize the reuse of the foundational model's capabilities across different tasks while still distinguishing the model's decisions for each task. This is a technical issue that cannot be solved by simply using methods like DETR. For example, as shown in Fig. 1 of the paper, an ID query for Prof. Geoffrey Hinton might get confused with broader information about the "person" category when using the traditional DETR decoder to process ID queries, instance queries, and semantic prototypes simultaneously. These technical challenges motivated us to propose the novel M-Former design.
> > > > > >
> > > > > > As shown in the table below, compared with recent method that directly adopt the DETR structure without considering task ambiguity, SINE achieves significant performance improvements.
> > > > > >
> > > > > > ||DAVIS|YouTube-VOS|
> > > > > > |---|---|---|
> > > > > > |DINOv [A] | 73.3|60.9|
> > > > > > |SINE|77.0|66.2|
> > > > > >
> > > > > > [A] Visual in-context prompting. CVPR 2024.
> > > > > >
> > > > > > Based on the considerations mentioned above, our insights are not incremental but provide valuable contributions to the research community.

---

> > > > > > > ### Author Response · Authors · 2024-08-14
> > > > > > >
> > > > > > > Dear reviewer,
> > > > > > >
> > > > > > > We sincerely thank you again for your great efforts in reviewing and discussing this paper. As the Author-Reviewer discussion period is going to end soon, we want to know whether the reviewer has any additional questions based on our previous response. Please don’t hesitate to let us know if you have any remaining concerns.
> > > > > > >
> > > > > > > Thank you,
> > > > > > >
> > > > > > > The authors

---

> > > > > > > ### Comment · Reviewer_2YmX · 2024-08-14
> > > > > > > **Re: Official Comment by Authors**
> > > > > > >
> > > > > > > Thanks for the authors for providing another round of rebuttal, which clearly addressed my major concerns **Q1** and **Q2**.
> > > > > > >
> > > > > > > However, I still hold my view that the novelties of this work from both theoratical perspective and representation learning respective are limited, especially for the loss design and the DERT based pipeline and the use of multiple cross-attention.
> > > > > > >
> > > > > > > Therefore, I decide to keep my initial rating, and could not vote acceptance.

---

> ### Author Response · Authors · 2024-08-14
>
> We believe that our response has answered the reviewer's concern. In addition, **the reviewer's response at the last minute of ddl is irresponsible**.

---

### Official Review · Reviewer_rhP3 · 2024-07-08

**Soundness:** 2
**Presentation:** 3
**Contribution:** 2
**Rating:** 5
**Confidence:** 3

**Summary:**

The paper proposes an image segmentation framework using in-context examples. To eliminate ambiguity from the in-context examples, multiple output masks are predicted. It uses a pre-trained image encoder to extract features from target and reference images, pools these features into ID and semantic tokens using the reference mask, and employs a Matching Transformer to decode the output masks. Experiments on various segmentation tasks demonstrate the effectiveness of the proposed method.

**Strengths:**

- The motivation on task ambiguity of segmentation with in-context example is clear.
- The paper evaluates the framework on few-shot semantic segmentation, few-shot instance segmentation, and video object segmentation on multiple datasets.

**Weaknesses:**

The novelty of the overall idea and network structure is limited. The idea to solve the ambiguity of prompts (in-context examples in this paper) by predicting multiple masks is from SAM. The network architectures are mainly based on DETR and Mask2Former. Though these choices are effective, they do not introduce innovations to the field. Overall, the paper shows solid practical utility,  and the limited novelty in its core idea and network design prevents me from giving it a higher rating. A stronger emphasis on introducing novel concepts or architectural advancements would enhance the impact and recognition of the work.

**Questions:**

N/A

---

> ### Author Rebuttal · Authors · 2024-08-05
>
> We thank you for your comments and the approval of our motivation and practical utility of SINE. We address your concerns here.
>
> ___
>
> >W1: The novelty of the overall idea and network structure is limited.
>
>
> To address the reviewer's concerns, we first discuss the contributions and academic insights of our paper. Then, we discuss the differences between SINE and other methods (SAM, DETR, and Mask2Former). Finally, we outline some perspectives. We hope these responses adequately address the issues and concerns raised.
>
> ## Contribution and Insight
>
> In-context segmentation is an important proxy task for unifying different segmentation tasks, and it has garnered significant attention from the research community. Our paper delves deeply into this task, and we summarize our contributions, novelty, and research insights as follows:
>
> 1. **Ambiguities in Visual Prompting**: This paper is the *first to explore the task ambiguity problem in prompts within in-context segmentation* (supported by **sZmi** and **nRMP**). This is an **important** and **novel** issue that has been **underexplored**. We investigate the conflicts among multiple tasks in in-context segmentation from the perspective of task ambiguity and *provide effective solutions*, offering valuable insights to the research community.
>
> 2. **Investigating the Capabilities of VFMs**: Utilizing Visual Foundation Models (VFMs) to address various tasks is becoming a research trend. Our paper explores how to efficiently transfer the visual representations of VFMs to in-context segmentation, a research paradigm that *currently lacks extensive exploration in the in-context segmentation field*.
>
> 3. **Lightweight and Effective Decoder**: We thoroughly analyze the challenges brought by task ambiguity (see Lines 181-185) and design the M-Former structure to address these issues. The novel dual-path design and Shared SA in M-Former ensure efficient decoding while avoiding information confusion across tasks of different granularity. Additionally, unlike previous methods, SINE achieves significant performance improvements with fewer trainable network parameters (19M). Thus, our method also demonstrates novelty.
>
>
> ## Differences between SINE and other methods
>
> Based on the above contributions, SINE fundamentally differs from SAM, DETR, and Mask2Former:
>
> **Comparison with SAM**
>
> - SAM is a promptable segmentation model that takes a point as input and outputs the corresponding mask. The ambiguity lies in the uncertainty of the segmentation granularity represented by the point. SAM addresses this by introducing different queries in the decoder to obtain masks of different granularities.
>
> - Resolving ambiguity in prompts (references) for in-context segmentation poses a more challenging task. In-context segmentation requires understanding the information in the prompt (e.g., category, position, shape) and learning the complex interaction relationships between the reference and target (approved by **sZmi**). Misunderstanding the prompt leads to incorrect outputs. Therefore, unlike SAM, SINE must learn the complex contextual relationships and avoid information confusion across different tasks.
>
> **Comparison with DETR and Mask2Former**
>
> - DETR and Mask2Former aim to train all model parameters on a specific dataset (e.g., COCO) to perform detection or segmentation on a limited number of categories.
>
> - SINE aims to effectively perform in-context segmentation in the open world by using the off-the-shelf VFMs. The parameters of the encoder are frozen. We have analyzed the difficulties in addressing task ambiguity in in-context segmentation, challenges that do not exist in the general detection and segmentation task. *Mask2Former and DETR are not designed for in-context segmentation, making it difficult to address these challenges.*
> The dual-path design and shared SA in M-Former are introduced specifically to address these challenges. As shown in Table 5(c), *compared to the Mask2Former decoder, M-Former effectively resolves task ambiguity, resulting in improvements across various tasks.*
>
> ## Perspective
>
> Finally, we want to convey our perspective to the reviewer: In the era of large models, we cannot only focus on network architecture design. Different models with the same architecture can have vastly different characteristics (e.g., DINOv2 shows excellent patch-level matching ability, while CLIP excels in image-text retrieval). We should also focus more on how to fully leverage the potential of pre-trained models with fewer parameters and computations. From this perspective, SINE provides valuable insights to the academic community.

---

> ### Comment · Reviewer_rhP3 · 2024-08-11
>
> Thank you to the author for the rebuttal. My concerns have been partially addressed.
> - "SINE must learn the complex contextual relationships and avoid information confusion across different tasks"
>
> Compared with SAM, the contextual information seems heavily dependent on ground truth reference masks, whereas SAM can progressively segment objects in multiple turns. Other segmentation models, like those from co-segmentation, only require image groups as inputs. These examples make the claim confusing to me.
>
> - "This paper is the first to explore the task ambiguity problem in prompts within in-context segmentation"
>
> I agree with the author that we cannot solely focus on network architecture design in the era of large models. However, the insights presented appear to be only incrementally novel, as they mainly build on multiple previous works.

---

> ### Author Response · Authors · 2024-08-12
>
> Thank you very much for your efforts and time in further discussing our work.
> Below, we provide a more detailed clarification regarding the remaining concerns.
>
> ___
>
> >Compared with SAM, the contextual information seems heavily dependent on ground truth reference masks, whereas SAM can progressively segment objects in multiple turns. Other segmentation models, like those from co-segmentation, only require image groups as inputs. These examples make the claim confusing to me.
>
> We will address the reviewer's question from the following two points. If our explanation is unclear, we would welcome further discussion.
>
> 1. *SINE is more cost-effective and broadly applicable.*
> When using SAM for processing a large number of images, it requires human interaction to segment each image, leading to significant labor costs. Co-segmentation requires ensuring that a group of images contains objects with the same semantic conception and cannot segment multiple objects with different semantics in complex scenes simultaneously. This limitation hinders its widespread application. In contrast, SINE only needs a single in-context example containing a reference image and mask to handle target images in different tasks without human intervention, making it more cost-effective and broadly applicable.
>
> 2. *Compared with SAM, why does SINE need contextual information.*
> For each image, SAM segments objects based on human interaction without learning semantics. This allows SAM to progressively segment objects in multiple turns, but its drawback is the need for human input for every image, leading to significant labor costs. In contrast, SINE can batch-process images using just one image and its mask, a practical advantage that SAM lacks. To achieve this, SINE needs to understand the contextual relationship between the reference image and target images. In fact, **SAM and SINE represent two vertically developed directions** of segmentation foundation models and can complement each other. For instance, in auto-labeling, SAM could label objects (e.g., a dog) in the first image, and SINE could use that image as an in-context example to label subsequent images, reducing costs.
>
> >I agree with the author that we cannot solely focus on network architecture design in the era of large models. However, the insights presented appear to be only incrementally novel, as they mainly build on multiple previous works.
>
> We appreciate the reviewer's agreement with our perspective. Below, we explain why our insights are not incremental:
>
> In widely used segmentation tasks such as semantic and instance segmentation, traditional methods require assigning a category label to the object, which becomes challenging when scaled to thousands of categories. *In-context segmentation overcomes these limitations by using visual concepts as annotations.* However, ambiguous prompts in in-context segmentation can lead to incorrect predictions, limiting its practical use. We are the first to consider and resolve task ambiguity in prompts for in-context segmentation, a significant contribution in both research and application, and we believe this challenge is inherently novel.
>
> In exploring the potential of DINOv2 for in-context segmentation, the challenge lies in *how to maximize the reuse of the foundational model's capabilities across different tasks while still distinguishing the model's decisions for each task*. This is a technical issue that **cannot be solved by simply using methods like DETR**. For example, as shown in Fig. 1 of the paper, an ID query for Prof. Geoffrey Hinton might get confused with broader information about the "person" category when using the traditional DETR decoder to process ID queries, instance queries, and semantic prototypes simultaneously. **These technical challenges motivated us to propose the novel M-Former design**:
>
> - Dual-Path Structure: M-Former's input includes queries and prototypes. Queries handle instance-level predictions, while prototypes, derived from the reference image, maintain semantic-level category information. SINE processes them separately, forming a dual-path structure to enable query learning and avoid interference from prototype's semantic-level information.
>
> - Shared SA: Queries include both ID and instance queries, representing different levels of granularity. To prevent interaction between ID queries and coarse-grained instance queries or prototypes, we introduce a well-designed attention mask (Fig. 2, top right) in the shared SA to effectively address this issue.
>
> As shown in the table below, compared with recent method that directly adopt the DETR structure without considering task ambiguity, SINE achieves significant performance improvements.
>
> ||DAVIS|YouTube-VOS|
> |---|---|---|
> |DINOv [A] | 73.3|60.9|
> |SINE|77.0|66.2|
>
> [A] Visual in-context prompting. CVPR 2024.
>
> Based on the considerations mentioned above, our insights are not incremental but provide valuable contributions to the research community.

---

> ### Comment · Reviewer_rhP3 · 2024-08-12
>
> Thank you to the author for their detailed analysis and explanation. In my opinion, the insights and novelty presented in the paper are limited, leading me to consider it as a borderline-to-rejection paper. I appreciate the extensive effort put into evaluating the proposed method. As a result, recognizing that the method is demonstrated as a strong baseline, I have adjusted my rating to borderline.

---

> > ### Author Response · Authors · 2024-08-12
> >
> > Thank you very much for your efforts and time again! This rebuttal and discussion will be helpful to improve our revised manuscript.

---

### Official Review · Reviewer_nRMP · 2024-07-10

**Soundness:** 3
**Presentation:** 3
**Contribution:** 3
**Rating:** 7
**Confidence:** 4

**Summary:**

The paper proposes a generalist model for image segmentation named SINE, which unifies multiple image segmentation tasks into the common formulation of visual in-context learning. This work aims to identify and model the task of object reidentification to reduce ambiguities within the in-context examples. By incorporating the modeling of each specific segmentation task within the SINE architecture, it effectively improves upon existing generalist models and achieves strong performance across a wide range of segmentation tasks.

**Strengths:**

1. This paper offers a valuable review of related works in in-context segmentation, analyzing the problems with the recent SegGPT and clarifying the differences of task setting between them.
2. The authors discuss the relationship between different segmentation tasks, effectively unifying them as instance segmentation, which is well-motivated.
3. This paper points out and addresses the ambiguities in visual prompting, which is currently an open research problem.
4. The authors conduct comprehensive experiments across various segmentation tasks, demonstrating significant performance improvements over recent generalist and specialist models.

**Weaknesses:**

1. This paper only addresses the ambiguity between instance and semantic segmentation. However, there is a broader ambiguity in visual prompts, such as spatial position, category, color, etc. The authors need to discuss these aspects in more detail.
2. Compared to SegGPT, SINE introduced additional Objects365 as extra training data. Although this was explained, it still seems to lack some fairness. Without using Objects365, can better performance be achieved than SegGPT? For example, by only using ADE20K, COCO, etc., and what is the performance on COCO-20i, PASCAL-5i?
3. Although the authors conducted numerous experiments, the impact of different backbones is missing. Since the backbone is frozen, different models might bring significant performance differences.

**Questions:**

1. The authors only conducted experiments on dinov2 vit-l. How do models of different sizes and different pre-training affect the results? For example, clip.
2. How does SINE perform few-shot learning? It seems that SINE can only accept a single reference image.

**Limitations:**

See weaknesses.

---

> ### Author Rebuttal · Authors · 2024-08-05
>
> We thank you for your comments and the approval of our motivation and performance. We address your concerns here.
>
> ___
>
> >W1: Discussion of more ambiguity in visual prompts.
>
> Thanks for your helpful suggestions. SINE is the first work to highlight task ambiguity in the visual prompts of in-context segmentation, initially focusing on resolving ambiguities among ID, instance, and semantic segmentation tasks (as these are more important and commonly used). We believe addressing ambiguities at this level is meaningful. For more complex ambiguities, such as full objects and parts, spatial position, category, and color, these can be addressed by incorporating multimodal in-context examples (e.g., image and text). We will add more discussions on this in the paper.
>
>
> >W2: Comparison of SINE and SegGPT by only using ADE20K, COCO.
>
> The table below compares the one-shot semantic segmentation results of training SINE using only ADE20K and COCO with SegGPT. SINE outperforms SegGPT on three benchmarks. Notably, SINE achieves 10% higher mIoU than SegGPT on LVIS-92i, indicating stronger class generalization capability in real-world image segmentation compared to SegGPT. Additionally, we are the first to explore the use of Objects365 in in-context segmentation. Table 5(a) of the paper shows that SINE's effective design leads to further improvements in generalization capability with the inclusion of Objects365.
>
> |  | COCO-20i | PASCAL-5i | LVIS-92i |
> | --- | --- | --- | --- |
> | SegGPT | 56.1 | 83.2 | 18.6 |
> | SINE | 67.1 | 86.3 | 28.8 |
>
>
> >W3,Q1: Impact of different backbones.
>
>
> We select DINOv2-S, DINOv2-B, DINOv2-L, and CLIP-L to explore the impact of different backbones. The conclusions are as follows:
>
> 1. DINOv2 Outperforms CLIP: DINOv2 achieves better performance than CLIP because it has general matching capabilities at both image and patch levels, allowing it to better understand complex contextual information between images. In contrast, CLIP captures image-text similarity, making it difficult to capture relationships between images, leading to poorer performance.
>
> 2. Larger DINOv2 Models Perform Better: Larger DINOv2 models have stronger representation capabilities, making it easier to capture contextual relationships, thus improving performance. This also indicates that SINE is scalable with the enhanced capabilities of the encoder.
>
>
>
> |  | COCO-20i | PASCAL-5i | LVIS-92i |
> | --- | --- | --- | --- |
> | SegGPT | 56.1 | 83.2 | 18.6 |
> | SINE DINOv2-S | 56.8 | 81.4 | 26.7 |
> | SINE DINOv2-B | 61.7 | 84.1 | 29.5 |
> | SINE DINOv2-L | 64.5 | 85.4 | 31.2 |
> | SINE CLIP-L | 34.8 | 57.3 | 16.1 |
>
> >Q2: How does SINE perform few-shot learning?
>
> Multiple reference image features and masks are concatenated in the spatial dimension. The resulting feature and mask can be treated as a single reference image, and the subsequent process remains the same.

---

> > ### Comment · Reviewer_nRMP · 2024-08-08
> >
> > The rebuttal has addressed my main concerns. Since the contribution of unifying segmentation tasks with in-context examples is clear, novel, well motivated, and well demonstrated, I would like to keep my original rating and recommend to accept this paper. I hope the authors can release their source code to benefit researchers in the same domain.

---

> > > ### Author Response · Authors · 2024-08-08
> > > **Reply**
> > >
> > > Thank you for the reviewer's recognition of our work. And we promise to open-source our code.

---

### Official Review · Reviewer_sZmi · 2024-07-15

**Soundness:** 2
**Presentation:** 2
**Contribution:** 3
**Rating:** 5
**Confidence:** 4

**Summary:**

SINE aims to resolve the problem of task ambiguity in in-context segmentation, where previous models struggled to accurately infer tasks based on in-context examples alone. This ambiguity arises because traditional models often fail to distinguish between different segmentation tasks like semantic segmentation, instance segmentation, or identifying specific objects based on the context provided. SINE utilizes a Transformer-based architecture where the encoder generates high-quality image representations and the decoder produces multiple task-specific output masks. In-context Interaction Module further enhances the encoder's output by establishing correlations between the target image and in-context examples, helping to better define segmentation tasks. Matching Transformer (M-Former), a dual-path Transformer decoder, updates object queries and semantic prototypes for precise task execution. It incorporates fixed matching and the Hungarian algorithm to resolve differences between tasks, ensuring that the output aligns well with the given task. SINE achieves impressive performance improvements over previous in-context segmentation models like SegGPT, particularly in handling multiple segmentation tasks simultaneously. It is shown to effectively address the issue of task ambiguity, producing relevant segmentation masks that are more aligned with the semantic content of the images.

**Strengths:**

[Experimental Results]. Extensive experiments across multiple benchmarks show significant improvements in multiple downstream tasks.

[Paper writing]. This paper is well-written and overall easy to follow.

[Interesting task formulation]. SINE was shown to effectively address the issue of task ambiguity, producing relevant segmentation masks that are more aligned with the semantic content of the images, which was not explored by the previous works in this domain.

**Weaknesses:**

[Concerns on Generalizability: Complex Interaction Relationships Beyond Semantically Similar Objects] While this paper adeptly handles in-context instance or semantic segmentation, numerous open-vocabulary segmentation models already capably segment novel classes or objects. A crucial demonstration of an in-context segmentation model's comprehension of in-context samples should extend beyond merely identifying semantically similar objects provided by context information. An intriguing task would involve, for example, segmenting objects situated on a table across multiple images—such as a bottle, a plate, and a book in successive images—and understanding whether SINE can recognize and segment another object on a table in a subsequent image. This requires the model to understand the interaction, the relationships, etc, that goes beyond segmenting semantically similar objects.

[Challenges with Complex In-Context Information] Often, a single pair of images suffices to provide the necessary in-context information reflecting the user's intent. However, more complex scenarios might require integrating multiple pairs of images to fully capture intricate in-context information. I am concerned about SINE's limitations in handling such complexity.

[Performance Discrepancy in Video Segmentation Compared to SegGPT] While SINE excels beyond SegGPT in instance segmentation on MSCOCO, it underperforms in video instance segmentation across most evaluated benchmarks. This result is concerning, considering video instance segmentation's unique demand for the model to sustain consistent correspondence across multiple video frames, a task more complex than segmenting static images. I am interested in understanding the specific reasons behind this performance gap.

[Unable to Handle Many Tasks SegGPT Supports]. SegGPT can actually support a lot more tasks, such as hierarchical in-context segmentation. I am concerned if SINE is overfitted on the instance/semantic segmentation tasks.

**Questions:**

Minor question: Why is the target image the same as the raw image in Figure 4a)?

Minor issues - typos: L307 "Results Table 7 compares ...." should be "Results Table 4 compares ...."? If it's not a typo, Table 4 was never discussed in the main paper.

**Limitations:**

Yes.

---

> ### Author Rebuttal · Authors · 2024-08-05
>
> We thank you for your comments and the approval of our task formulation. We address your concerns here.
> ___
>
> >W1: Concerns on Generalizability.
>
> Fig. 1 in the attached PDF shows SINE's capability in handling complex interaction relationships.
> - In Fig. 1(a), the reference consists of multiple images, each containing different objects (box, cup, keyboard, mouse). When using these as in-context examples, SINE can segment one or more semantically different objects on a desk.
> - In Fig. 1(b), with a reference containing only one object, the in-context example cannot represent complex interactions, and thus no segmentation result is provided.
> - In Fig. 1(c), replacing multiple single-object images with a single image containing multiple objects yields the same effective results.
>
> These experiments indicate that SINE has the potential to handle complex interaction relationships beyond semantically similar objects.
>
>
> >W2: Challenges with Complex In-Context Information.
>
> We believe SINE can handle complex in-context infromation because the contextual relationships between complex scenarios can be captured more efficiently in the representation space of DINOv2.
> SINE leverages this characteristic of DINOv2 to accurately understand and fully capture intricate in-context information in multiple pairs of images. Extensive experiments show that SINE performs better in the few-shot setting than the one-shot setting, indicating its ability to effectively comprehend and capture in-context information from multiple samples. Figure 1(a) in the attached PDF also shows SINE's capability to understand and handle the complex relationships in in-context examples.
>
>
> >W3: Performance Discrepancy in Video Segmentation Compared to SegGPT.
>
> We think the reason behind this performance gap is that SegGPT trains all model parameters (300M), while SINE uses a **simpler in-context fusion module** and **fewer learnable parameters (19M)**. The detailed analysis is as follows:
>
> SegGPT uses a broader dataset, concatenates references and targets spatially, and employs an ViT architecture. It trains all model parameters (300M), with self-attention effectively capturing relationships between video frames.
>
> SINE aims to learn a general and lightweight decoder to efficiently transfer representations from pre-trained DINOv2 (encoders) to in-context segmentation. The frozen DINOv2 is limited to capture in-context information. For efficiency, SINE deploys only a simple in-context fusion module (1.58M) to learn in-context relationships, limiting its ability to handle inter-frame relations in complex videos.
>
> Considering image tasks or simple videos (e.g., DAVIS), we believe that the overall performance of SINE are satisfactory. In paiticular, compared to recent generalist segmentation models SEEM[A] and DINOv[B] (which train all parameters), SINE's efficient design shows greater potential in video tasks (see the table below).
>
> Although SINE currently has limitations in learning complex inter-frame relationships in videos, we believe that by designing more suitable In-Context Interaction module, the current paradigm holds greater potential for solving in-context segmentation tasks. We will explore this further in future work. These discussions will be added to the paper.
>
> ||DAVIS|YouTube-VOS|
> |---|---|---|
> |SegGPT|75.6| 74.7|
> |SEEM [A] |58.9| 50.0|
> |DINOv [B] | 73.3|60.9|
> |SINE|77.0|66.2|
>
>
> [A] Segment everything everywhere all at once. NIPS 2023.
>
> [B] Visual in-context prompting. CVPR 2024.
>
> >W4: Unable to Handle Many Tasks SegGPT Supports.
>
> SINE and SegGPT have different motivations.
> - SegGPT aims to verify visual in-context learning can unify different segmentation tasks by using a broader range of datasets, such as semantic segmentation (ADE20K), instance segmentation (COCO), and part segmentation (PACO). Hence SegGPT can perform hierarchical in-context segmentation.
> - SINE aims to resolve the task ambiguity in prompts. As the first work to highlight this problem, we initially focus on resolving ambiguities among ID, instance, and semantic segmentation tasks (as these are more important and commonly used).
>
> When PACO is included as training data, **SINE can perform part segmentation like SegGPT**, as shown in Fig.2 of the attached PDF. This shows that our method does not overfit instance/semantic segmentation.  Additionally, the results in Table 1 of the paper (LVIS-92i) and Fig.3 of the attached PDF indicate that SINE has stronger class generalization capability compared to SegGPT in real-world image segmentation.
>
>
> >Q1: Why is the target image the same as the raw image in Figure 4a ?
>
> In Fig. 4(a), the giraffe demo is selected from two video frames with a large temporal gap. The reference is from an earlier frame with only one giraffe, while the subsequent frame gradually introduces another giraffe. This demo tests whether SINE can identify objects with the same ID without temporal information.
>
>
> >Q2: typos: L307 "Results Table 7 compares ...."
>
> Thanks for pointing this out. It should be Table 4. We will fix this error in the paper.

---

> > ### Comment · Reviewer_sZmi · 2024-08-12
> >
> > I would like to thank the authors for their responses. Most of my questions have been addressed. Therefore, I will maintain my current rating as borderline accept.

---

> > > ### Author Response · Authors · 2024-08-13
> > >
> > > Thank you for your feedback and your great efforts. Any further questions/suggestions would be also appreciated.

---

### Author Rebuttal · Authors · 2024-08-05

# **General Response**

We thank the reviewers for recognizing that our paper points out an open research problem, i.e., the ambiguities in visual prompting (**nRMP**). The motivation on task ambiguity is clear (**rhP3**), and we effectively address task ambiguity (**sZmi,nRMP**). Our method is rationale and effective (**2YmX**), significantly improving performance across various segmentation tasks (**sZmi,nRMP**), and demonstrating solid practical utility (**rhP3**).

In addition, we will carefully address the issues and suggestions raised by the reviewers and will make further revisions and improvements to our paper. In every Official Review of the reviewers below, we have provided responses to the questions and suggestions made by the reviewers, and we hope these responses adequately address the issues and concerns raised. **If the reviewers have any further questions regarding our paper and responses, please let us know.**
___

In the General Response, we highlight the contributions and insights of this work.

In-context segmentation is an important proxy task for unifying different segmentation tasks, and it has garnered significant attention from the research community. Our paper delves deeply into this task, and we summarize our **contributions**, **novelty**, and academic **insights** as follows:

1. **Ambiguities in Visual Prompting**: This paper is the *first to explore the task ambiguity problem in prompts within in-context segmentation* (supported by **sZmi** and **nRMP**). This is an **important** and **novel** issue that has been **underexplored**. We investigate the conflicts among multiple tasks in in-context segmentation from the perspective of task ambiguity and *provide effective solutions*, offering valuable insights to the research community.

2. **Investigating the Capabilities of VFMs**: Utilizing Visual Foundation Models (VFMs) to address various tasks is becoming a research trend. Our paper explores how to efficiently transfer the visual representations of VFMs to in-context segmentation, a research paradigm that *currently lacks extensive exploration in the in-context segmentation field*.

3. **Lightweight and Effective Decoder**: We thoroughly analyze the challenges brought by task ambiguity (see Lines 181-185) and design the M-Former structure to address these issues. The novel dual-path design and Shared SA in M-Former ensure efficient decoding while avoiding information confusion across tasks of different granularity. Additionally, unlike previous methods, SINE achieves significant performance improvements with fewer trainable network parameters. Thus, our method also demonstrates novelty.

Taken together, these contributions represent novel insights to the academic community.

___

The attached PDF contains 3 additional results:

1. Generalizability of SINE beyond semantically similar objects. (**sZmi**)

2. Visualization of part segmentation. (**sZmi**)

3. Visualization comparisons between SINE and SegGPT on video and image tasks. (**2YmX**)

---

### Decision · Program_Chairs · 2024-09-25

**Decision:**

Accept (poster)

**Comment:**

This paper received split recommendations after the post-rebuttal discussion. The rebuttal adequately addressed most reviewer concerns about technical correctness and experimental validation. During an extensive author-reviewer discussion, the authors provided additional detailed analyses and explanations that satisfied three reviewers. Only Reviewer 2YmX recommended rejection on the grounds of insufficient technical novelty. All reviewers agreed on the clarity of the motivation, the quality of writing and presentation, and the adequacy of the experimental validation. Despite the lack of technical breakthroughs, the proposed solution to the problem of task ambiguity in in-context segmentation seems ingenious and effective enough to interest the broad NeurIPS community.